

**Reframing gullies as recharge zones in dryland landscapes of the Loess Plateau, China**
Zhenxia Ji[a,b,c,d], Alan D. Ziegler[e], Li Wang[a,b,c,d]
[a] College of Natural Resources and Environment, Northwest A&F University, Yangling 712100, China
[b] College of Soil and Water Conservation Science and Engineering, Northwest A&F University, Yangling,
712100, China
[c] State Key Laboratory of Soil and Water Conservation and Desertification Control, Northwest A&F
University, Yangling 712100, China
[d] Institute of Soil and Water Conservation, Chinese Academy of Sciences and the Ministry of Water
Resources, Yangling 712100, China
[e] Andaman Coastal Station for Research and Development, Kasetsart University, Ranong, Thailand
Correspondence to: Li Wang (wangli5208@nwsuaf.edu.cn)
State Key Laboratory of Soil and Water Conservation and Desertification Control, Institute of Soil and
Water Conservation, Northwest A&F University.
Address: Xinong Road, #26, Yangling, Shaanxi Province 712100, China





**Abstract**

Large gullies in dryland landscapes are often viewed as indicators of land degradation, yet in some settings they may serve critical ecohydrological functions—supporting groundwater recharge and subsurface connectivity. In China's Loess Plateau, we assess these functions in the Nianzhuang Catchment using a multi-indicator approach that integrates stable isotopes ($\delta^2$H, $\delta^{18}$O), chloride concentrations, and groundwater level fluctuations. Our results show that precipitation is the dominant source of recharge for shallow pore water within gully zones, while deeper fissure water is replenished more slowly through percolation from the upper layers. Restoration interventions—particularly check dams and ponds—act as focal points for groundwater infiltration, enhancing recharge in otherwise limited dryland systems. Estimated annual recharge (238–241 mm) accounts for over 43% of annual precipitation, far exceeding typical rates observed in nearby tableland and hilly areas. These findings revise prevailing assumptions by positioning gullies not simply as degraded features, but as hydrologically active zones that can buffer seasonal variability and support ecosystem resilience. The study advances a conceptual framework for using isotopic damping, chloride accumulation, and recharge partitioning as indicators of landscape function in semi-arid regions, offering valuable tools for dryland monitoring and restoration planning.

**Keywords:** surface water, spring water, pore water, fissure water, connectivity

## 1. Introduction

Groundwater recharge is a critical yet poorly understood component of hydrological cycles in dryland catchments. It is shaped by the precipitation regime, surface landcover heterogeneity, the integrity of the subsurface regolith, the characteristics of the underlying bedrock, and human interventions (Vries and Simmers, 2002; Owuor et al., 2016; Salek et al., 2018; Xu and Beekman, 2019; Zhang et al., 2020; Li et al, 2024; Medici et al., 2024). Although favorable subsurface flow pathways can locally enhance recharge, dryland regions—situated along climatic ecotones and shaped by complex land–atmosphere–biosphere feedbacks—are highly sensitive to even modest shifts in water availability. Small changes in soil moisture or runoff routing can cascade across catchments are various scales, amplifying existing vulnerabilities to ecological and social systems (Nicholson, 2011; Huang et al., 2017; Berg et al., 2016). In these "fragile" and diverse landscapes, understanding the processes that govern



when, where, and how groundwater is replenished—including the countervailing influences of
vegetation dynamics, geomorphology, and engineered features—is essential for sustaining ecosystems,
securing water resources, and informing land restoration and catchment management (Gleeson et al.,
2016; Jasechko and Perrone, 2021; Scanlon et al., 2006).

Despite a growing body of research on groundwater recharge in (semi-)arid regions, significant

knowledge gaps remain in landscapes with pronounced spatial heterogeneity—such as slopes, hilltops,
and gullies—where infiltration pathways and recharge processes can diverge sharply over short distances
(Tooth, 2012; Manna et al., 2018; Letz et al., 2021). Among these landforms, gully systems—often
regarded as hallmarks of land degradation—may paradoxically serve as focal points for recharge,
capturing and infiltrating surface runoff during episodic rainfall events (Tan et al., 2017; Li et al., 2024;
Xue et al., 2025). Notably, gully systems may facilitate the rapid downslope transport of contaminants
such as agricultural contaminants and sediments (Lian et al., 2025; Qu et al., 2025). However, the role
of gullies in promoting vertical infiltration into groundwater is highly dependent on local subsurface
connectivity and permeability conditions. Moreover, their broader hydrological functions remain poorly
quantified—especially under the influence of widespread human interventions such as check dams and
artificial ponds. While these structures are typically designed to arrest land degradation, they can
substantially alter surface–subsurface connectivity and reshape recharge dynamics (Lamontagne et al.,
2021; Wang et al., 2023).

This study focuses on groundwater recharge in the gully systems in loess soils. Worldwide, loess

covers approximately 6% of the land surface area, forming discontinuous east–west belts in the mid-
latitude forest-steppe, steppe, and desert-steppe zones of both hemispheres (Liu, 1985; Pécsi, 1990; Li et
al., 2020). Among these, the Chinese Loess Plateau accounts for approximately 7.4% of the global loess
area (635,280 km²; Li et al., 2020). The setting for our investigation, semi-arid landscape has been shaped
by severe soil erosion, extensively modified by engineered landforms; and it is now characterized by
chronic water scarcity (Fu et al., 1999; Liu et al., 2017; Liu and Li, 2017; Li et al., 2021). In such
vulnerable environments, understanding the sources and sustainability of groundwater recharge is critical
for long-term water resource management (Ajjur and Baalousha, 2021; Meles et al., 2024). Groundwater
is a lifeline for rural communities in the hilly–gully region, yet scientific attention has largely bypassed
the gullies themselves. Most research has centered on recharge processes in tablelands and loess-covered
hills (Huang et al., 2011; Li et al., 2017; Lu, 2020; Wang et al., 2024), leaving the hydrological role of



gully systems—despite their striking prominence in the landscape—largely in the shadows (Liu et al.,

2011).

In this study, we integrate stable isotope analysis ($\delta^2$H and $\delta^{18}$O), chloride concentration

measurements, water table fluctuation estimations, and hydro-statistical modeling to do the following: (i)
quantify pore water recharge rates; and (ii) trace flow paths among surface water, pore water, and fissure
water. This integrated approach aims to advance understanding of groundwater dynamics in complex
dryland terrains, generating process-based insights critical for sustainable water and land management
in gully-dominated systems—not only across the Loess Plateau, but in drylands globally.

**2. Hydro-geomorphological processes of the Loess Plateau**

The Loess Plateau in China stands out as one of the most ecologically and hydrologically distinctive

landscapes in the world (Fu et al., 2017). Spanning over 640,000 km², it harbors the planet's largest and
deepest loess deposits and has long served as a cradle of Chinese civilization (Li et al., 2021). The Loess
Plateau is also one of the most severely eroded regions in the world, shaped by the interplay of highly
erodible soils, intense summer storms, and a long history of farming on sloping lands (Shi and Shao,
2020). For centuries, steep hillslopes were cultivated without adequate soil conservation, removing
vegetation and exposing loess soils to heavy runoff during short, high-intensity monsoonal rains. These
conditions led to widespread gully formation—hallmarks of degradation tightly linked to land use and
rainfall extremes (Wang et al., 2006; Fu et al., 2011; Jin et al., 2020). The altered hydrological cycle on
the plateau has led to sharp declines in both streamflow and groundwater levels, resulting in acute water
scarcity across this arid to semi-arid region (200–750 mm annual rainfall) (Liang et al., 2015; Wang et
al., 2023; Chen et al., 2023). Understanding hydrological processes in this disturbed setting is essential
for guiding soil conservation, optimizing groundwater recharge, and ensuring the long-term
sustainability of water resources for both people and ecosystems.

The plateau's complex stratigraphy—characterized by loess layers that can reach depths of up to

350 meters and average around 90 meters, with generally low permeability—governs groundwater
storage, recharge processes, and subsurface flow behavior (Qiao et al., 2017; Zhu et al., 2018). The region
is further divided into a range of distinctive landforms, or subregions, also shaped by variable erosion
processes (Yang et al., 2009): Loess Yuan (flat, high-elevation tablelands); Loess Liang (elongated
ridges); Loess Mao (rounded or oval-shaped hills); hills; and large-scale gullies (Fig. A1). While the



Loess Yuan remains relatively unincised by rivers/streams and maintains a smooth surface morphology,
the Liang and Mao formations exhibit profound dissection—shaped predominantly by fluvial erosion
and hillslope processes, respectively.

Hills in the region exhibit undulating terrain shaped by prolonged weathering and diffuse surface

runoff. In contrast, gully systems reflect both long-term geomorphic evolution and more recent
intensification driven by human activity—making them products of millennial-scale natural processes
and modern land-use pressures (Li et al., 2021; Jia et al., 2024). Prominent gully systems, which can
stretch for several kilometers, serve as primary conduits for concentrated runoff, facilitating the
downslope transfer of water, sediment, and eroded material. In doing so, they reflect and reinforce
ongoing landscape evolution and contribute to downstream sedimentation challenges (Zhu et al., 2018).

To combat degradation in the hilly–gully region, extensive afforestation efforts have been

implemented on hillslopes, while hydraulic structures—including check dams and ponds—have been
constructed in gullies (Feng et al., 2016; Xue et al., 2025). These interventions have significantly altered
runoff dynamics and water budgets (Huang et al., 2013; Yuan et al., 2022). While afforestation has
reduced soil moisture in hilly areas via enhanced evapotranspiration (~8.7%), hydraulic engineering has
increased soil moisture in gullies by 21% (Wang et al., 2019, 2020; Zhao et al., 2019, 2024). Notably,
the additional water retained within gully systems offsets an estimated 44% of the water loss from
afforested hillslopes, partially reshaping the local water cycle (He et al., 2020; Zhao et al., 2024). Under
such intensive human modification, the Plateau's inherently complex landforms and stratigraphy have
come to exert even greater control over groundwater recharge dynamics (Li et al., 2024).

Research on groundwater recharge in the Loess Plateau's gully regions is incomplete, with most

studies concentrating on deep profiles in tableland and hilly areas (Huang et al., 2011; Li et al., 2017; Lu,
2021; Wang et al., 2024). Early scholars proposed that fissures and caves in loess enabled preferential
flow, allowing precipitation to reach groundwater (Yan and Wang, 1983). However, Xu et al. (1993)
argued that the vertical fractures in the loess layer do not facilitate continuous water movement; instead,
the vertical joints and large pores may act as barriers, which is caused by air-blocking effects. Li (2001)
argued that the formation of dried soil layers further disrupts groundwater recharge pathways, causing
precipitation to cycle within the soil–plant–atmosphere system rather than contributing meaningfully to
groundwater recharge. In recent years, researchers have applied isotope tracing, hydrochemical analysis,
and model simulations to study recharge mechanisms in deep-loess regions (Lu, 2020; Xiang, 2020; Shi





et al., 2021; Wang et al., 2023). These studies suggest that recharge occurs primarily through slow piston
flow, with precipitation infiltrating thick soil profiles, slowly recharging groundwater in a process that
can take decades to hundreds of years (Huang et al., 2013; Tan et al., 2017; Li et al., 2024). Piston flow
refers to a type of water movement through the unsaturated (vadose) zone or an aquifer where newly
infiltrating water pushes the existing water downward, much like a piston in a cylinder (Gee and Hillel,

1988).

However, deep-profile recharge mechanisms observed in tableland and hilly areas may not apply to

gully landscapes on the plateau. Regional-scale analyses of 40 years of soil moisture data show that
precipitation infiltration in thick loess deposits is typically restricted to shallow depths, even though loess
thickness in tableland and hilly areas ranges from 56.5 to 204.5 m (Wang et al., 2024; Qiao et al., 2017).
In contrast, loess in gully regions is generally much thinner—often less than 50 m (Zhu et al., 2018).
Infiltration on loess slopes appears similarly limited (Fig. 1).

During field observations in the 2023 rainy season in the Nianzhuang Catchment, located in the

hilly–gully region of the Loess Plateau, we found little evidence of preferential flow through cracks or
macropores. Instead, infiltration appeared slow and driven predominantly percolation through the matrix
(Wang et al., 2024). That year, total rainfall from May to October amounted to 420 mm, with 115 mm
falling in September alone. Consistent with earlier studies (Xu et al., 1993; Li, 2001), only a few surface
cracks showed signs of infiltration, and even then, the water was absorbed by surrounding soils and failed
to infiltrate deeper (Fig. 1c). Moreover, soil profiles remained unsaturated from the surface to deeper
layers, indicating that precipitation infiltration is generally insufficient to recharge groundwater (Qiao et
al., 2017). After a 41-mm rainfall event occurring over four days, infiltration depths reached only 20–30
cm at the top of the slope, compared with 80 cm at mid-slope positions (Fig. 1b, c).

These patterns indicate that infiltration is limited at higher slope elevations, with much of the water

moving laterally downslope as overland flow and accumulating in gully areas, where conditions are more
conducive to groundwater recharge. Previous studies have shown that gullies and other topographic
depressions function as key recharge zones, enabling concentrated surface flows to infiltrate more deeply
and contribute to subsurface water stores (Gates et al., 2011; Liu et al., 2011; Zhao et al., 2021). Building
on this foundation, advancing our understanding of gully-driven recharge requires targeted investigation
of interactions among precipitation, surface water, and groundwater—particularly the flow paths and
magnitudes of near-surface infiltration. These insights are critical for guiding water resource



management and ecological restoration across the Loess Plateau.

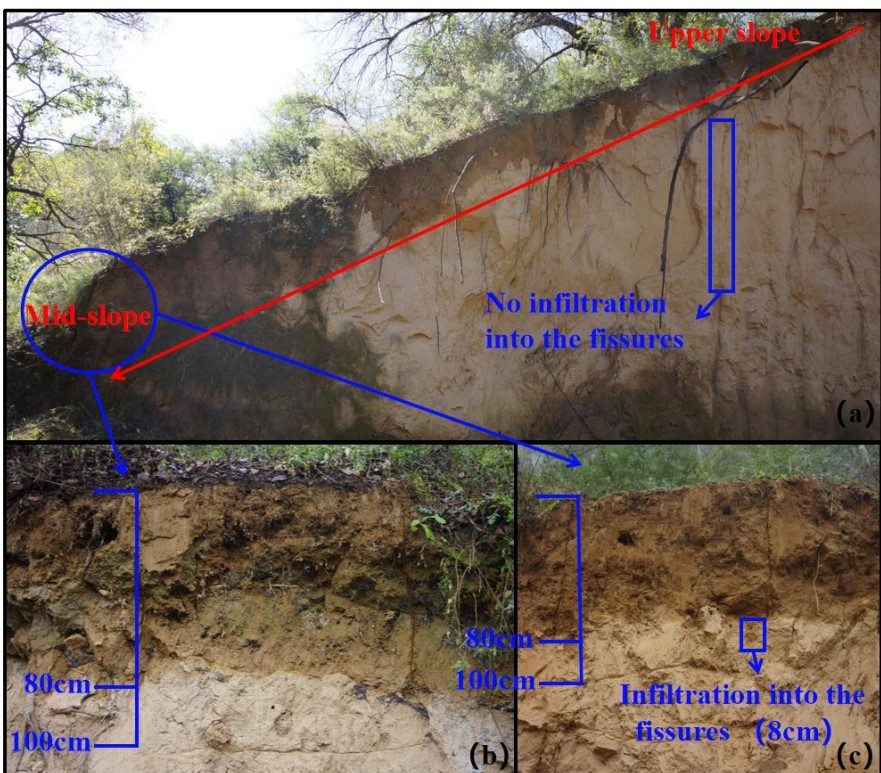


Fig. 1. The topographic profile of the Nianzhuang catchment in the hilly region of the Loess Plateau. Full
profile from the top to mid-slope (a); two repeated mid-slope profiles (b, c). The photo was taken after a
41 mm rainfall event over four days. Subsequent measurements showed that infiltration depths reached
only 20–30 cm at the top of the slope, compared to approximately 80 cm at the mid-slope positions.

**3. Sampling site**
The Nianzhuang Catchment is located northwest of Yan'an City in Shaanxi Province, China
(approximately 36°42′N, 109°31′E). As a tributary of the Yanhe River—which ultimately flows into the
Yellow River—the catchment spans 53.94 km² and includes the well-studied Yangjuangou sub-
catchment (3.11 km²; ~36°35′N, 109°32′E), previously investigated in numerous hydrological and
ecological studies (Fu et al., 1999; Liu and Li, 2017). Elevation ranges from 896 to 1,269 m, with terrain
gradually sloping from northwest to southeast (Fig. 2). The region experiences a semi-arid continental
monsoon climate, with a mean annual precipitation of approximately 550 ± 100 mm, concentrated





between July and September (Liu et al., 2017).

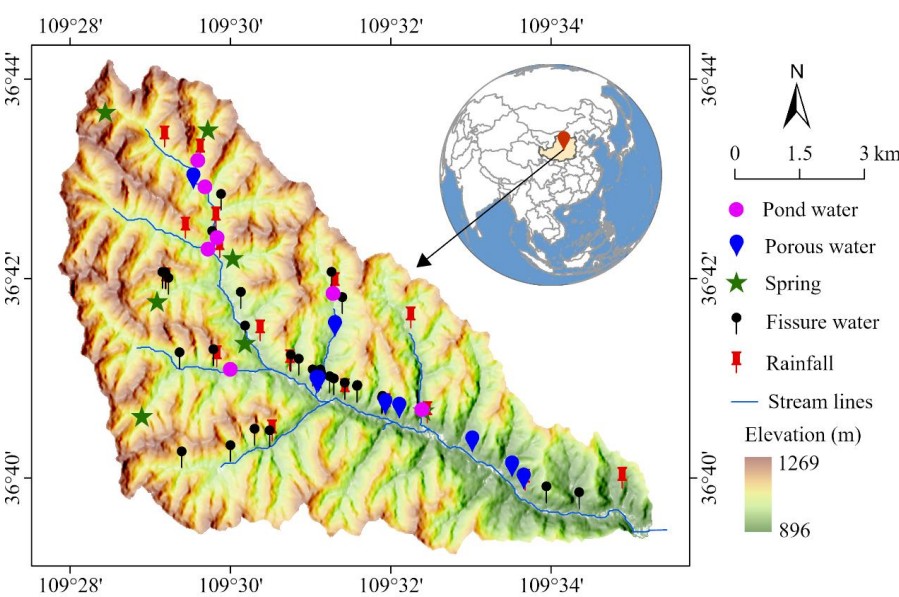

Fig. 2. The geographical location and sampling sites for rainfall, pond water, pore water, spring water,
and fissure water in the Nianzhuang catchment. The Nianzhuang catchment is located in the hilly and
gully region of the central Loess Plateau, with elevations ranging from 869 to 1269 m. The average depth
of pore water wells is $8.0 \pm 1.5$ m (range: 4–10 m), while that of fissure water wells is $57.6 \pm 29.2$ m
(range: 25–170 m). These sampling sites represent locations where both rainy and dry season samples
were collected, and are all situated within the gully areas of the catchment.

The catchment features highly dissected loess terrain, with characteristic soils and landforms such

as Loess Liang (ridges), Loess Mao (mounds), and steep loess slopes (Cai et al., 2019). Gullies—often
"V"- or "U"-shaped—dominate the lower-lying regions and serve as important recharge zones. These
landforms, together with ancient landslides, minor collapses, and sinkholes, highlight the geomorphic
instability of the Loess Plateau landscape (Li et al., 2021).

The stratigraphy of the catchment reflects the typical layered structure of the Loess Plateau, which

plays a key role in controlling groundwater recharge. In upland hilly areas, thick loess deposits overlie
bedrock, with the Upper Pleistocene Malan Loess—light grayish-yellow, loosely textured, and silt-rich
(>60%)—characterized by well-developed vertical joints and abundant hematite and goethite. Beneath
it lies the Middle Pleistocene Lishi Loess, a grayish-yellow to light brown unit with prominent jointing
and higher iron mineral content. Below the loess, the Neogene Red Clay appears as a distinctly reddish,





calcareous nodule–bearing aquitard due to its low permeability. The entire sequence rests on Jurassic
sandstone–conglomerate bedrock, composed mainly of quartz-rich fluvial–lacustrine deposits.
Loess thickness in the Liang and Mao regions often exceeds 150 meters, resulting in deep water
tables and limited groundwater accessibility. In contrast, gully zones exhibit distinctly different
hydrogeological characteristics. Here, thinner loess layers overlie Neogene and Jurassic formations,
sometimes interbedded with coal seams up to 5 meters thick (Fig. 3a–c). The significant reduction in
loess thickness—combined with the relatively high permeability of Neogene coarse sandstone and
conglomerate (7.5–36.19 m/d)—creates favorable conditions for infiltration and focused recharge.
These dynamics are especially evident at gully heads, where surface runoff from adjacent uplands
converges and infiltrates, forming efficient recharge zones. As a result, gully areas tend to have shallower
water tables and more rapid water renewal, making them more suitable for domestic groundwater use.
Springs frequently emerge at gully bottoms where lateral flow is facilitated at the loess–bedrock interface.
Streams in this dry environment are largely intermittent.

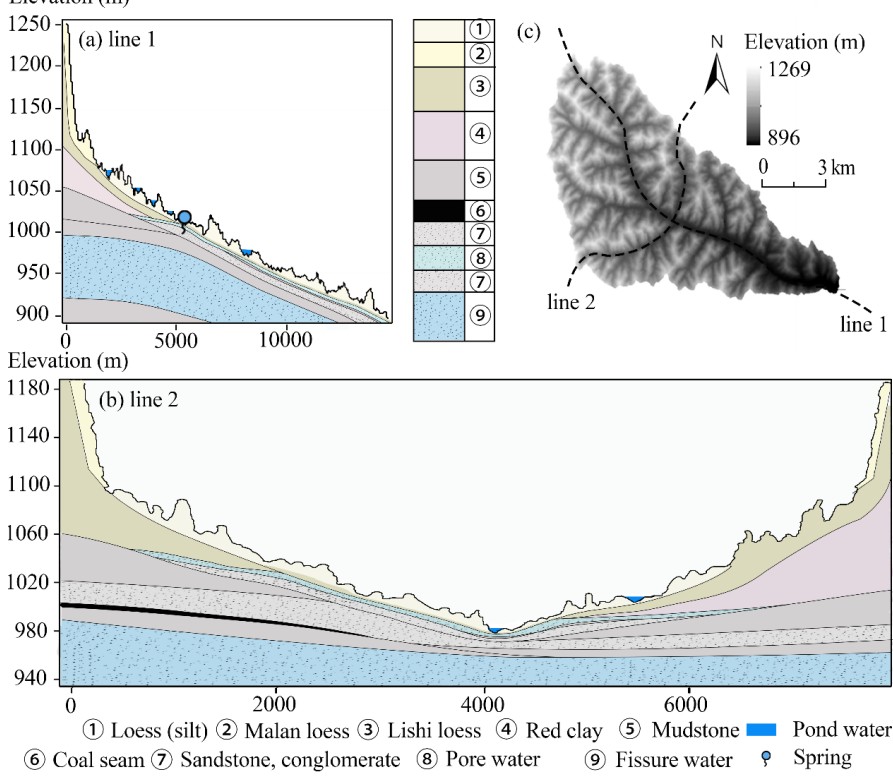

① Loess (silt) ② Malan loess ③ Lishi loess ④ Red clay ⑤ Mudstone ▮ Pond water
⑥ Coal seam ⑦ Sandstone, conglomerate ⑧ Pore water ⑨ Fissure water ⚲ Spring

Fig. 3. Hydrogeologic cross-section of the study area. Cross-section along Line 1 (Northwest-Southeast)





(a); cross-section along Line 2 (Southwest-Northeast) (b); location map of Line 1 and Line 2 within the study area (c). The Malan Loess (11.7–12.6 Ka BP) and Lishi Loess (12.6–78.1Ka BP) are two major Quaternary loess stratigraphic units in China. Based on hydrogeological research, the stratigraphy of the hilly region features a multi-layer structure from top to bottom: Upper Pleistocene Malan Loess, Middle Pleistocene Lishi Loess, Neogene Red Clay and Mudstone (*2.58–23.03* Ma BP), and Jurassic Sandstone and Conglomerate (145–201.3 Ma BP). In the gully region, the stratigraphy includes Holocene loess (silt, 11.7 ka BP–present), Middle Pleistocene Lishi Loess, Neogene sandstone and mudstone, and Jurassic sandstone and conglomerate, with some areas containing coal seams up to 5 meters thick.

Groundwater in the catchment can be broadly categorized into three types: pore water, spring water, and fissure water. Pore water is stored in permeable sandstone and conglomerate aquifers beneath loess and above mudstone or red clay. Conceptually, "pore water" here refers to groundwater in a saturated aquifer, not to soil moisture. Fissure water occurs within fractured bedrock aquifers but is spatially discontinuous due to irregular fissure development. Hydraulic conductivity in these sandstone and conglomerate aquifers ranges from 0.0218 to 0.471 m/day (Cai et al., 2019). Spring water emerges primarily at gully bases—especially in upper catchments—and originates from both pore and fissure sources, possibly supplemented by surface or pond water.

Over recent decades, landscape rehabilitation through the Grain for Green Project and land reshaping under the Gully Land Consolidation Project have significantly altered the hydrological regime (Fu et al., 1999; Liu et al., 2017). Historically, surface runoff in the degraded catchment was flashy and episodic due to sparse vegetation. However, ecological restoration and small-scale engineering interventions—such as check dams, terraces, roads, and ponds—have moderated surface hydrology. Surface runoff, generated primarily during storm events, now contributes alongside delayed baseflow from groundwater recharge and interflow. The latter is often limited by the thick unsaturated zone in upland loess areas but may be enhanced in gully regions, where stratigraphy and land use favor infiltration (Wang et al., 2024; Gates et al., 2011). Gully areas also contain numerous check dams and ponds, with most water sourced from Hortonian overland flow and direct rainfall. These small water bodies, often constructed for erosion control and water retention, influence local hydrological dynamics and may play a role in enhancing infiltration and recharge.

**4. Methods**



Our approach integrates stable isotope analysis (δ²H and δ¹⁸O), chloride concentration analysis, and
water table fluctuation monitoring to investigate groundwater recharge dynamics. The isotopic
composition of water bodies reflects both their origins and the processes they undergo—such as
evaporation, infiltration, and mixing (Wan and Liu, 2016; Kumar et al., 2019; Dasgupta et al., 2024).
Precipitation, surface water, and groundwater typically exhibit distinct isotopic signatures due to these
differing pathways (Gleeson et al., 2016; Kuang et al., 2019; Al-Oqaili et al., 2020). When isotopic
patterns among water sources converge, it often indicates strong hydrological connectivity (Yang and
Wang, 2023). Because stable isotopes behave conservatively, they serve as effective tracers of water
sources and flow paths (Gleeson et al., 2016; Al-Oqaili et al., 2020; Dasgupta et al., 2024). In parallel,
water table fluctuation (WTF) monitoring provides a means of estimating recharge by observing changes
in groundwater levels in response to precipitation events (Nachabe, 2002; Heppner and Nimmo, 2005;
Gumuła-Kawęcka et al., 2022). By combining these complementary methods, this study aims to elucidate
groundwater recharge pathways and quantify recharge rates in gully regions—thereby identifying key
recharge zones and advancing our understanding of groundwater processes in the Loess Plateau.

**4.1. Field measurements of hydrological data**
Precipitation was collected from October 24, 2023, to October 24, 2024, using a weather station
situated in an open field within the catchment. Continuous groundwater level data were recorded from
September 24, 2023, to December 20, 2024. Groundwater pressure and temperature were monitored
using Onset HOBO U20-001-03 sensors (20 m range), and water table levels were calculated based on
the measured pressure data. The conversion relationship between water pressure and groundwater level
is given by $Y = 0.86 \times X - 22.1$ where $Y$ represents the groundwater level and $X$ represents the water
pressure. Notably, the monitoring well is located in the pore water layer of the gully region. The well is
hand-dug (1.1 m wide, 10 m deep) and is unaffected by human activities.
Soil physical properties were assessed using the cutting ring method, based on undisturbed soil cores
collected at five depth intervals: 0–10, 10–20, 20–30, 30–40, and 40–50 cm. Quadruplicate samples were
taken near groundwater monitoring wells in the gully using pre-weighed cutting cylinders. The samples
were immediately transported to the laboratory for analysis. Bulk density, capillary porosity, non-
capillary porosity, total porosity, and field water capacity were determined following the LY/T 1215-
1999 standard for forest soil water-physical properties. Soil particle size distribution was analyzed using



a laser particle size analyzer at the College of Natural Resources and Environment, Northwest A&F
University. Soil texture classification followed the USDA system: sand (0.05–2 mm), silt (0.002–0.05
mm), and clay (<0.002 mm) (Dane et al., 2002).

**4.2 Water sampling**
A total of 181 water samples were collected from various locations in rainy season (September 2023,
99 samples) and dry season (April 2024, 82 samples); see Fig. 2. Rainy season samples included 48 from
rainfall, 7 from pond water (water retention reservoirs), 9 from spring water, 9 from pore water, and 26
from fissure water. During the dry season, samples included 31 from rainfall, 6 from pond water, 8 from
pore water, 29 from fissure water, and 8 from spring water.
Pore water was collected from several shallow, hand-dug wells measuring approximately 1.1 meters
in diameter and 4–10 meters in depth. Fissure water was sampled from deeper, narrow-diameter wells
(0.2 meters wide, 25–170 meters deep). In areas with numerous deep wells, we employed random
sampling to ensure representative coverage of fissure water sources. To minimize the risk of collecting
stagnant water, all pore and fissure water wells were purged for 10–15 minutes prior to sampling. Spring
water was collected directly from natural discharge points, although most springs in the region exhibit
low flow rates—typically less than 0.1 L/s, occasionally reaching up to 0.2 L/s.`
A total of 18 bulk rainfall collectors were randomly and evenly distributed across the 54 km$^2$ study
area, and samples were collected immediately following rainfall events. For nighttime precipitation,
samples were collected the next morning at 6:00 AM. During the study period, we collected two types of
precipitation samples: (1) spatial samples from individual events (18 in the rainy season and 15 in the
dry season) across the catchment, capturing spatial variability; and (2) sequential events at a fixed station
(30 in the rainy season and 16 in the dry season), characterizing seasonal inputs.
During sampling, 100 mL collection bottles were rinsed two to three times with the sample water,
then slowly filled to minimize air exposure. After filling, the bottles were tightly sealed with screw caps
and further secured with Parafilm to prevent evaporation and contamination. All samples were
immediately stored in a portable cooler at 4°C and transported to the laboratory for isotopic and chloride
concentration analysis.

**4.3. Isotopic analysis**





The δ²H and δ¹⁸O values of the water samples were determined using a Los Gatos Research liquid
water isotope analyzer (Model 912-0032, LGR Inc., California, USA) at the Institute of Water-Saving
Agriculture in Arid Areas of China, Northwest A&F University. Each sample was injected six times in
the following sequence: three standard injections, followed by six natural sample injections, and then
three additional standard injections. The isotope ratios were calculated using the average composition
from injections 4 through 6.
Isotope values are expressed in delta (δ) notation, which represents the relative difference in isotope
ratio between a sample and the Vienna Standard Mean Ocean Water (VSMOW) reference. The
measurement precision was ±0.5‰ for δ²H and ±0.1‰ for δ¹⁸O. The delta values were calculated using
the following equations:
$$\delta^{18}O = \left(\frac{R_{sample}}{R_{standard}}\right) - 1 \tag{1}$$
$$\delta^{2}H = \left(\frac{R_{sample}}{R_{standard}}\right) - 1 \tag{2}$$
where $R_{sample}$ and $R_{standard}$ are the ratios of heavy to light isotopes ($^{18}O/^{16}O$ or $^{2}H/^{1}H$) in the sample
and the standard, respectively. Results are expressed in per mil (‰).

**4.4. Mixing process of different water bodies**

Inverse transit time proxies (ITTPs) were calculated to assess differences in water transit times and
mixing processes across various water bodies (Tetzlaff et al., 2009). ITTPs are defined as the ratio of the
standard deviation of δ¹⁸O in the water sample (e.g., pond water, spring water, pore water, or fissure
water) to that in precipitation over the same time period:
$$\mathbf{ITTP} = \frac{\sigma\delta^{18}O(sample)}{\sigma\delta^{18}O(precipitation)} \tag{3}$$
This ratio captures the attenuation of seasonal isotopic variability in δ¹⁸O as water moves through
the landscape. In general, ITTP values less than 1 indicate substantial damping of the precipitation
signal—consistent with longer water residence times, greater mixing, and larger storage volumes.
Conversely, values approaching 1 suggest minimal damping and rapid flow paths.
However, interpretation of ITTPs must also account for fractionation processes. In particular,
evapotranspiration (ET) selectively removes lighter isotopes ($^{16}O$), enriching the remaining water in
heavier isotopes ($^{18}O$). This enrichment can artificially increase the variance of δ¹⁸O in near-surface or
shallow soil water compartments, inflating ITTP values even in systems with relatively slow transit times



(Tetzlaff et al., 2009). This is especially relevant in arid and semi-arid regions, where ET can dominate
the water balance during dry seasons.

**4.5. Hydraulic connectivity estimation**

Structural Equation Modeling (SEM) has been widely applied in water science to evaluate complex

causal relationships among hydrological, geological, and anthropogenic variables—particularly in
studies of groundwater contamination and water quality degradation (Wu, 2010; Lupi et al., 2019; Xie et
al., 2025). In this study, we adapted SEM as an exploratory framework to assess hypothesized recharge
linkages among water sources, using dual-isotope ($\delta^2$H–$\delta^{18}$O) data from rainfall, pond water, spring water,
pore water, and fissure water. Although SEM is not a mass-conserving approach and is less commonly
used in isotope-based flowpath analysis, it enables estimation of statistically significant relationships and
indirect linkages within a hypothesized recharge system.

Given the potential for isotopic signatures to be altered by evaporation, mixing, or other non-

conservative processes, results must be interpreted with caution. Pathways with p-values > 0.05 were
excluded during model refinement, and the final model met standard goodness-of-fit criteria (degrees of
freedom < 3, RMSEA < 0.05, CFI > 0.95, NFI > 0.95). SEM analysis was conducted using SPSS Amos
26.0 (IBM SPSS, Chicago, Illinois, USA).

In addition, we applied variance partitioning to evaluate the relative contributions of different water

sources to pore and fissure water. This method decomposes the total variance in isotopic composition
into components attributable to individual sources (e.g., precipitation, pond water, spring water), offering
a complementary estimate of source influence. While useful, this approach remains subject to the same
limitations as SEM—particularly the challenges of isotopic overlap and limited resolution in
environments affected by mixing and evaporation (Lai et al., 2022).

To further constrain recharge pathways, we incorporated chloride ion (Cl⁻) as a conservative tracer.

Unlike stable isotopes, chloride is unaffected by evaporation or biological processes, making it a robust
indicator for identifying recharge sources and tracking subsurface water movement. Chloride
concentrations in all water samples were analyzed using an ion chromatograph (DIONEX ICS-1100) at
the College of Natural Resources and Environment, Northwest A&F University, China. Each sample was
analyzed in triplicate, with charge balance errors maintained below 5% to ensure analytical accuracy.
This rigorous approach enhances the reliability of chloride data, supporting its integration with isotopic





indicators in source attribution.

**4.6. Groundwater recharge**
Groundwater recharge in the gully zone is quantified using the water table fluctuation (WTF)
method, which infers recharge and discharge events from temporal changes in groundwater levels (Healy
and Cook, 2002; Gumuła-Kawęcka et al., 2022). We recognize that recharge can originate from three
hydrological sources: (1) surface water; (2) the unsaturated zone (3) and the saturated zone (Scanlon et
al., 2022; Wang et al., 2024). Among these, estimates based on the saturated zone are generally most
reliable, as recharge from the unsaturated zone reflects potential inputs that may never reach the water
table (Beven and Germann, 2013; Huang et al., 2019). The WTF method is widely used for estimating
saturated zone recharge due to its high temporal resolution and conceptual simplicity (Xu et al., 2024).
Based on previous site-specific studies (Wang et al., 2024), this method is well-suited for our analysis.
The water table fluctuation method assumes that changes in the groundwater table result solely from
recharge or discharge, assuming a constant specific yield ($Sy$) over time (Healy and Cook, 2002; Obuobie
et al., 2012). The formula is as follows:
$R_i = S_y \frac{\Delta H_i}{\Delta t}$                                              (4)
where, $R$ is the groundwater recharge (mm), $Sy$ is the specific yield of the aquifer, $\Delta H_i$ (where
$\Delta H_i > 0$) is the groundwater table rise caused by recharge between day $i-1$ and $i$, and $t$ is time period.
Specific yield, which represents the proportion of water that drains freely from the saturated zone under
gravity, was determined using two methods: the soil texture empirical method and the test pit method
(Liang, 2016).
Empirical values for soil texture are referenced in Table A1. The test pit method for estimating $S_y$ is
described as follows:
$S_y = TP - FWC$                                              (5)
where, $TP$ (total porosity) and $FWC$ (field water capacity) were measured using the cutting ring
method.
We applied two methods to calculate the daily groundwater table increments ($\Delta H_i$). The RISE
method assumes that recharge occurs only when the groundwater table elevation increases between two
consecutive days (Gumuła-Kawęcka et al., 2022). Thus, $\Delta H_i = H_i - H_{i-1}$ if $H_i > H_{i-1}$, otherwise,



$\Delta H_i = 0$. The master recession curve (MRC) method assumes that, in the absence of recharge, the
groundwater table declines daily by a specific amount ($\Delta H_{MRCi}$). This amount represents a simplified
approximation of discharge processes in the aquifer, particularly lateral outflow to nearby surface water
bodies. MRC establishes a functional relationship between a daily decrement of the water table ($\Delta H_{MRCi}$)
and the water table elevation ($H_{i-1}$) during periods without recharge.
$$\Delta H_{MRCi} = A \cdot H_{i-1} + B \tag{6}$$
where, the coefficients A and B were fitted for each piezometer based on data from periods of
continuous groundwater table decreases lasting longer than two weeks. The daily water table increment
due to recharge was then calculated as: $\Delta H_i = H_i - H_{i-1} + \Delta H_{MRCi}$ if $H_i > (H_{i-1} - \Delta H_{MRCi})$,
otherwise, $\Delta H = 0$.
Notably, we used water dynamics from October 24, 2023, to October 24, 2024, to calculate pore
water recharge, as this period exhibited clear groundwater fluctuations, making it more representative.

**5. Results**
**5.1. Soil properties**
The upper 50 cm of the soil profile is composed primarily of silt ($64.6 \pm 0.6\%$), with smaller but
nearly equal proportions of clay ($18.0 \pm 1.3\%$) and sand ($17.4 \pm 1.6\%$), classifying the loess as silt loam
according to the International Union of Soil Sciences (IUSS) scheme (Fig. 4a). Total porosity and field
water capacity decreased slightly with depth, averaging $24.5 \pm 1.9\%$ and $21.3 \pm 1.7\%$, respectively (Fig.
4b). Specific yield remained relatively consistent within the 10–50 cm depth interval, averaging
$3.2 \pm 0.8\%$--falling within the expected empirical range for silt loam soils (2%–7%) (Fig. 4c).

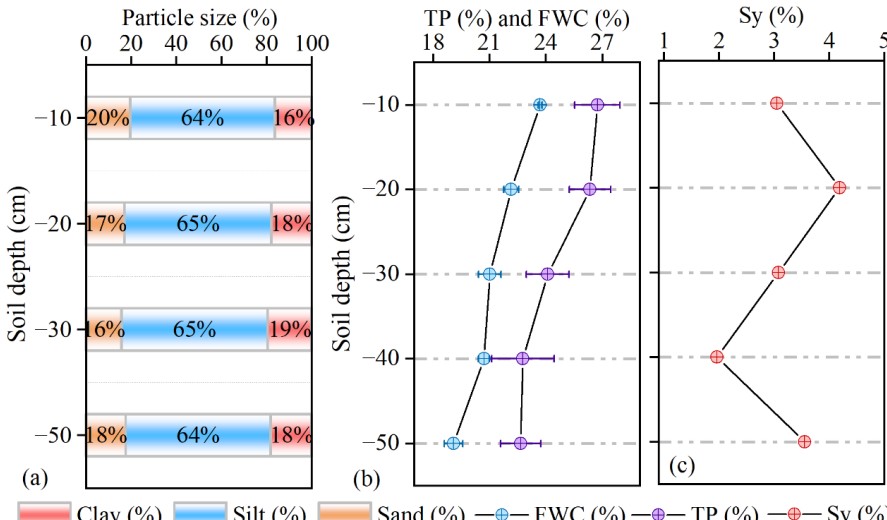

Fig. 4. Vertical variation in soil texture and water retention characteristics in the gully region of the Loess Plateau. (a) Soil particle size distribution by depth, showing relatively uniform composition across layers (10–50 cm), dominated by silt (64–65%), with moderate clay (16–20%) and low sand (16–20%) content. This fine-textured profile supports high moisture retention and slows infiltration, promoting delayed recharge. (b) Depth profiles of total porosity (TP) and field water capacity (FWC) reveal increases with depth to 40 cm, with FWC reaching ~27%, suggesting greater water-holding capacity in subsoil layers and enhanced buffering of infiltrated water. (c) Specific yield (Sy) peaks at −20 cm (4.5%) but decreases with depth, indicating shallower layers are more responsive to infiltration and release, while deeper layers tend to store water with minimal drainage. Collectively, these physical properties reflect a vertically stratified soil system where near-surface layers regulate infiltration pulses, and deeper layers act as long-term storage, shaping the timing and magnitude of subsurface recharge.

## 5.2. Hydrological signatures of rainfall, surface water, and groundwater sources

Pond water and rainfall exhibit similar spatial isotopic patterns, with more positive $\delta^2H$ values ($\delta^2H > -55$‰) than spring water, pore water, and fissure water (Fig. 5a, b). These values are line with the notion of direct rainfall and Hortonian runoff are the primary source of pond water. In contrast, the $\delta^2H$ values of pore, spring, and fissure water show little seasonal variation and are consistently more negative ($\delta^2H < -55$‰), than mean rainfall, indicating longer residence times and reduced evaporative influence.



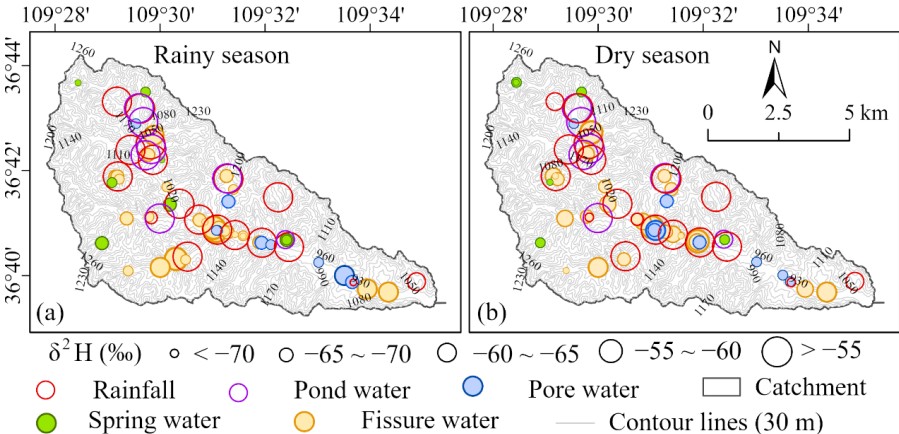

Fig. 5. The spatial distributions of $\delta^2H$ values during the (a) rainy season and (b) dry season for rainfall, pond water, spring, pore water, and fissure water in the gully region of the Loess Plateau. To highlight spatial differences among water sources, $\delta^2H$ values were classified into five intervals: $< -70‰$, $-70$ to $-65‰$, $-65$ to $-60‰$, $-60$ to $-55‰$, and $> -55‰$. Sampling points are color-coded by water type: red for rainfall, purple for pond water, blue for pore water, green for spring water, and orange for fissure water.

The $\delta^2H$–$\delta^{18}O$ relationships and box plots for each water source reveal key insights into the dominant hydrological processes occurring during the rainy season (Fig. 6). Firstly, rainfall follows a Local Meteoric Water Line (LMWL) of $\delta^2H = 7.7·\delta^{18}O + 8.9$ ($R^2 = 0.95$), which is closely aligned—though slightly offset—from the Global Meteoric Water Line (GMWL: $\delta^2H = 8·\delta^{18}O + 10$) (rainy season; Fig. 6a, c). This alignment confirms that precipitation in the region has a typical meteoric origin. Additionally, minimal evaporative enrichment occurred prior to collection. The relatively wide interquartile range of rainfall $\delta^{18}O$ values suggests that precipitation was derived from storm systems with considerable isotopic variability, reflecting differences in rainfall intensity, air mass origin, and temperature (Kumar et al., 2019; Oqaili et al., 2020; Dasgupta et al., 2024). However, this variability remains moderate compared with global patterns that span extreme rainfall events and broader climatic gradients.

Pond water, in contrast, exhibits a clear evaporation signature with a shallower slope of $\delta^2H = 5.6·\delta^{18}O − 17.1$ (rainy season; $R^2 = 0.95$), $\delta^2H = 4.6·\delta^{18}O − 20.7$ (dry season; $R^2 = 0.74$). This signature aligns with expectations for surface water bodies, where open exposure facilitates fractionation. The box plot confirms strong evaporative enrichment, with median values shifted significantly toward more



positive $\delta^{18}$O and $\delta^2$H compared to rainfall (Fig. 6a, c). Pond water maintains a relatively consistent slope
and range across seasons, reinforcing its stable evaporative signature and less dynamic recharge behavior.

Spring water shows a clear seasonal transition in its isotopic composition. In the rainy season, its

evaporation line ($\delta^2$H = 6.2·$\delta^{18}$O – 11.4; $R^2$ = 0.75) falls closer to the LMWL, suggesting that spring
discharge is augmented by recent rainfall, likely delivered through rapid infiltration and shallow
subsurface flow pathways during high-intensity events (Fig. 6a, b). However, the isotopic values of
spring water are substantially more depleted than those of precipitation, indicating that older water stored
in the porous subsurface aquifer dominates the overall spring flow composition composed of new and
relatively old water.

During the dry season, the isotopic slope flattens and deviates further from the Local Meteoric Water

Line (LMWL), reflecting increased evaporative influence or prolonged residence times (Fig. 6c, d). This
seasonal shift suggests that as rainfall inputs decline, spring discharge becomes increasingly composed
of slow-draining, older water that has undergone greater isotopic modification—either through mixing
or evaporation in near-surface storage zones. Collectively, these patterns suggest that spring water acts
as a dynamic integrator of recharge processes—rapidly responding to event-driven infiltration during the
rainy season, yet also reflecting the delayed mobilization of older water stored in the subsurface. This
behavior may be partly explained by a piston-like displacement mechanism, where incoming rainfall
pushes pre-existing groundwater toward discharge zones.

Pore and fissure water show remarkably similar isotopic signatures during the rainy season. Pore

water, again sampled from a porous subsurface aquifer, follows a fitted line of $\delta^2$H = 4.3·$\delta^{18}$O – 27.9
(rainy season; $R^2$ = 0.74), while fissure water, likely drawing from the same aquifer but through
weathered bedrock pathways, fits $\delta^2$H = 3.7·$\delta^{18}$O – 32.0 (rainy season; $R^2$ = 0.70). These slopes are
significantly flatter than those of rainfall, pond, or spring water, a pattern interpreted as evidence of
evaporation prior to recharge or mixing with evaporated surface water. However, the box plots of the
isotope data present a different picture. Both pore and fissure waters are systematically more depleted in
$\delta^{18}$O and $\delta^2$H than precipitation, and their narrow interquartile ranges suggest a relatively uniform
isotopic composition (Fig. 6a-d). Rather than showing the enrichment expected from evaporation, these
depleted and stable values point toward recharge dominated by a limited number of isotopically light
rainfall events, such as early-season storms or high-altitude convective systems.



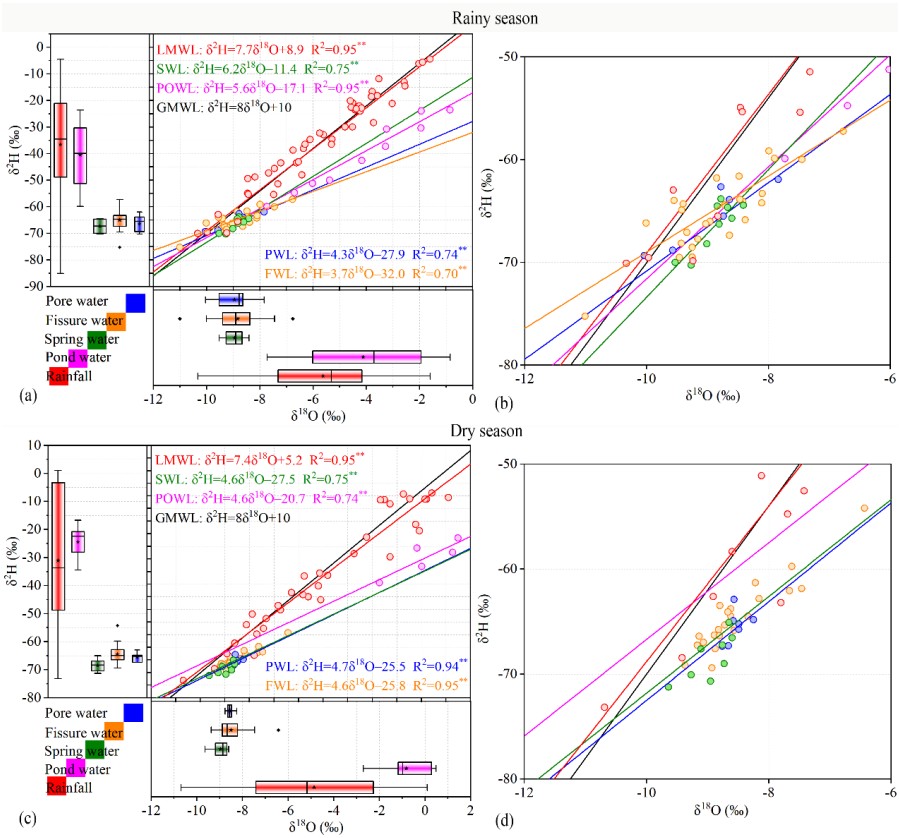

Fig. 6. Dual stable isotopic compositions of rainfall, pond water, spring water, pore water, and fissure water during the (a) rainy season and (b) dry season in the gully region of the Loess Plateau. The black line represents the global meteoric water line (GMWL, $\delta2H=10 + 8\delta18O$). GMWL is the global meteoric water line of Craig, LMWL is the local meteoric water line, SWL is the spring water line, POWL is the pond water line, FWL is the fissure water line, and PWL is the pore water line. Panels (b) and (d) are magnified views of (a) and (c), respectively, highlighting the isotopic compositions of pore water, fissure water, and spring water (x-axis: –12 to –6‰; y-axis: –80 to –50‰).

This apparent contradiction—flat regression slopes alongside depleted and uniform isotopic compositions—can be reconciled by considering longer residence times and mixing in the subsurface. The aquifer likely functions as a hydrological buffer, smoothing short-term isotopic variability and maintaining a signature reflective of older recharge. In this context, the low slopes may not indicate evaporation but instead reflect the dampening of seasonal isotope fluctuations through storage and subsurface mixing (Table A2).




Complimenting the isotope data, Cl⁻ levels in pore water consistently fall between those of
precipitation and pond water across both seasons (Fig. A2), supporting a mixed recharge origin. This
trend aligns with the isotopic evidence from the rainy season and supports the interpretation that pond
water contributes to pore water recharge via vertical percolation through the vadose zone, particularly
during high-rainfall periods when infiltration capacity is exceeded. The lack of a similar isotopic pattern
in the dry season likely reflects stronger evaporative enrichment of pond water, which masks its potential
contribution to pore water recharge in dual-isotope space.
The inverse transit time proxies (ITTPs) broadly support the dual-isotope interpretations of water
source dynamics. Pond water exhibited the highest ITTP values ($1.5 \pm 0.7$), indicating rapid turnover and
limited subsurface storage. These elevated values likely reflect inputs from direct rainfall and overland
flow, as well as evaporative enrichment, which increases isotopic variability and can artificially shorten
the apparent residence time. In contrast, pore water ($0.7 \pm 0.3$) and fissure water ($0.6 \pm 0.5$) showed lower
ITTPs, consistent with longer residence times, greater subsurface mixing, and attenuation of seasonal
isotopic signals due to delayed recharge. Spring water had the lowest ITTPs ($0.3 \pm 0.2$), reflecting slow
subsurface transport and integration of older water sources. While these patterns align with conceptual
expectations of residence time and flow path length, the limited number of samples—particularly for
pond, spring, and pore water—warrants caution in interpreting seasonal dynamics (Fig. 7).

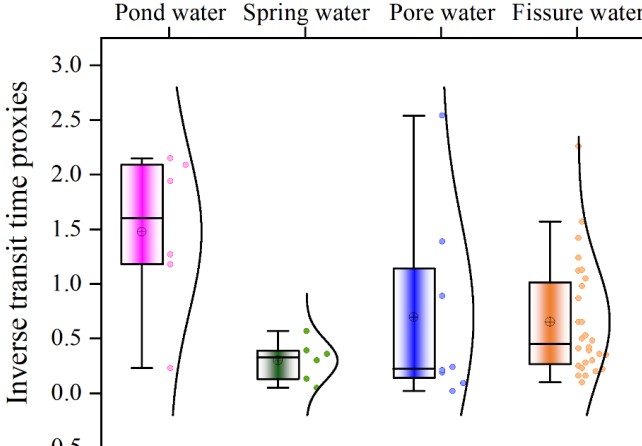


Fig. 7. Boxplots and kernel density estimates of inverse transit time proxies (ITTPs) for pond water,
spring water, pore water, and fissure water. Higher ITTP values indicate shorter water transit times since
precipitation, while lower values suggest longer residence and greater isotopic damping. Pond water





exhibited the highest and most consistent ITTPs (median ≈1.5), implying rapid recharge from recent
rainfall or stormflow. Spring water showed the lowest ITTPs (≈0.3), consistent with longer subsurface
flow paths and storage. Pore and fissure water displayed intermediate and more variable ITTPs, reflecting
mixing between recent and older water sources, as well as seasonal differences in infiltration and soil
moisture replenishment.

**5.3. Hydrological linkages and recharge efficiency**
The SEM analysis reveals significant hydrological linkages among different water bodies in the
catchment, with particularly well-defined pathways connecting rainfall, pond water, pore water, and
fissure water (Fig. 8a, b). Several key pathways identified in the model are supported by multiple lines
of observational evidence, including isotopic composition, chloride concentrations, and water age (ITTP).
Rainfall contributes over 73% to pore water recharge, far exceeding the <17% contribution from pond
water (Fig. 8c).
However, the SEM results indicate that the total effect of pond water on pore water is stronger than
that of rainfall (Fig. 8b). In SEM, the total effect includes both direct pathways (e.g., pond water → pore
water) and indirect pathways mediated by other variables (e.g., rainfall → pond water → pore water).
This apparent contradiction likely stems from the strong statistical association between rainfall and pond
water, as pond water is primarily derived from rainfall and shares similar isotopic signatures. As a result,
the model may overestimate pond water's influence on pore water due to overlapping signals and
correlated pathways.
These findings underscore the importance of integrating multiple lines of evidence rather than
relying solely on SEM outputs. For example, chloride concentrations in pore water more closely resemble
those of pond water, suggesting mixed recharge from both sources and highlighting the potential for pond
water to play a prominent role under certain spatial or temporal conditions (Fig. 2a). Although dry-season
isotopic data provide limited support for a strong pond water–pore water connection, the spatial
distribution of chloride offers compelling evidence that pond water can contribute significantly to pore
water recharge, particularly in localized areas or during specific recharge events.
At deeper levels, the linkage between pore water and fissure water is supported by their nearly
identical isotope values and similar ITTPs, suggesting a shared subsurface origin and a strong
hydrological connection (Fig. 8a, b). In contrast, although there is some hydrological connectivity

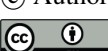



between pore water and spring water, differences in isotopic slopes and residence times may lead to an
overestimation of their interaction. However, their nearly identical chloride concentrations provide more
direct and reliable evidence of connectivity.

Although the model results initially suggest that rainfall—mediated through pond water—is the

primary source of pore water recharge, discrepancies among the different indicators call for a more
critical interpretation of the evidence. The contradictions observed across isotopic, chloride, and ITTP
data underscore the need for further quantitative validation.

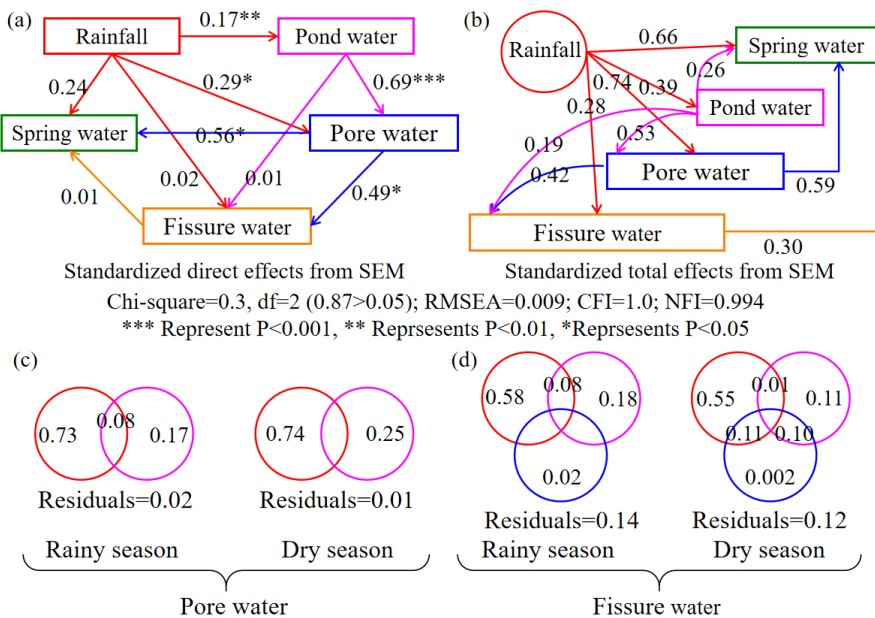


Fig. 8. Structural equation modeling (SEM) and variance partitioning results illustrating hydraulic
connectivity among water sources in the gully region of the Loess Plateau. Panels (a) and (b) show the
standardized direct (a) and total effects (b) among rainfall, pond water, pore water, spring water, and
fissure water, based on $\delta^{18}O$ and $\delta^{2}H$ data. Arrows indicate hypothesized water flow pathways, with line
thickness proportional to effect size. Asterisks denote statistical significance (*$P < 0.05$, **$P < 0.01$,
***$P < 0.001$). The model fit is excellent ($\chi^2 = 0.3$, df $= 2$, RMSEA $= 0.009$, CFI $= 1.0$, NFI $= 0.994$),
supporting the robustness of these inferred connections. Panels (c) and (d) present variance partitioning
results showing the relative contributions of source waters to pore water and fissure water during the
rainy and dry seasons, respectively. In panel (c), rainfall (red) and pond water (pink) explain a large
portion of pore water variability, with some shared explanatory power and modest residuals. In panel (d),



fissure water reflects a more complex origin, with contributions from rainfall (red), pond water (pink),
and pore water (blue), and greater overlap and residuals, especially during the dry season.

To address the contradictions observed in the SEM and variance partitioning results, we apply the

water table fluctuation method to independently estimate the recharge rate from rainfall to pore water.
Groundwater level fluctuations in the gully system revealed clear seasonal recharge dynamics, with an
initial rise in the pore water table beginning on October 24, 2023, followed by a decline through early
spring (March 1, 2024) and a gradual recovery starting June 20, 2024 (Fig. 9a). Between October 24,
2023, and October 24, 2024, cumulative recharge was estimated at $238.0 \pm 6.0$ mm (RISE) and
$241.4 \pm 6.0$ mm (MRC), with 159 and 167 recharge days, respectively (Fig. 9b). Given that annual
precipitation totaled approximately 550 mm, recharge efficiency was approximately 43–44%,
underscoring the significance of focused infiltration in sustaining shallow aquifer recharge within the
gully environment.

This recharge efficiency is lower than the precipitation-to-pore water contribution estimated by the

variance decomposition method and may more accurately reflect "actual" recharge, as statistical
estimates can be biased by similarities in isotopic signatures. When integrated with dual-isotope and
SEM analyses, the WTF-based results support a conceptual model in which storm-driven runoff is
efficiently captured and redistributed through loess soil matrices and retention structures (e.g., ponds,
check dams), activating a hierarchy of shallow and deeper subsurface flowpaths. These flowpaths link
pore water, fissure water, and spring discharge across the complex gully landscape, reflecting both
vertical and lateral connectivity within the groundwater system.

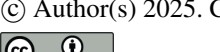

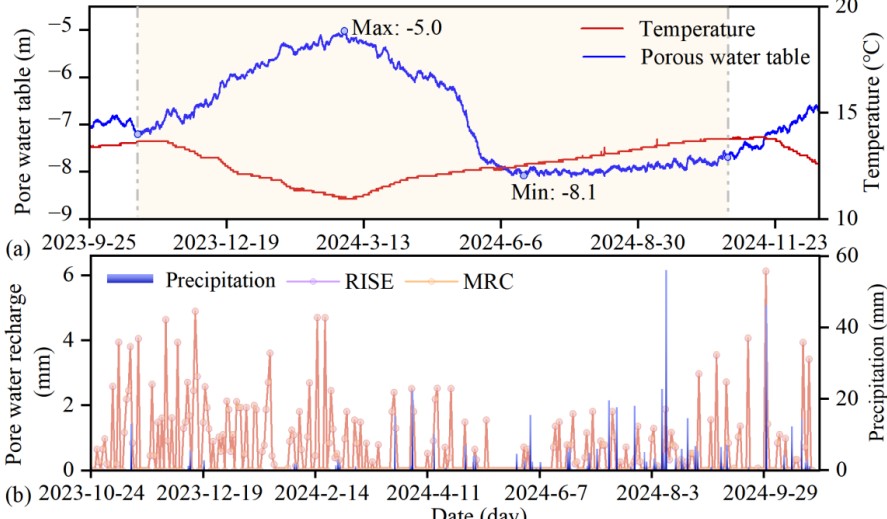


Fig. 9. Temporal dynamics of pore water table depth, temperature, precipitation, and recharge in the gully

region of the Loess Plateau. (a) Daily time series of pore water table depth (blue line) and surface

temperature (red line) from September 2023 to November 2024. The water table fluctuates seasonally,

rising from ~−8.1 m in late summer to a maximum of ~−5.0 m in early spring (March 2024), indicating

delayed infiltration and cool-season recharge. (b) Daily precipitation (blue bars) and modeled pore water

recharge estimates using the RISE and MRC methods. Most recharge events occur from October to April,

even when rainfall is not especially high, while warm-season precipitation contributes little to recharge—

likely due to increased evaporative losses and shallow soil retention. Together, these patterns suggest

strong seasonal control on recharge processes, with effective infiltration primarily occurring during

cooler, low-evaporation periods.

## 6. Discussion

### 6.1. Isotopic compositions of various water bodies in the gully region

In hydrological studies, the isotopic composition of water bodies reflects both sources and changes

in hydrological processes (Wan and Liu, 2016; Kumar et al., 2019; Dasgupta et al., 2024). Precipitation,

surface water, and groundwater usually exhibit differences in isotopic characteristics due to variations in

evaporation, infiltration, and mixing (Gleeson et al., 2016; Kuang et al., 2019; Al-Oqaili et al., 2020).

However, similar isotopic distributions among different water bodies often indicate a strong hydrological



connection in their water sources (Yang and Wang, 2023).
Our study found that rainfall and pond water have similar spatial isotopic distribution patterns,
indicating that pond water primarily originates from rainfall. This reflects the region's geographical and
hydrological characteristics. In this severely eroded gully region, the local government has constructed
an extensive network of ponds and check dams to capture hillside runoff (Liu et al., 2017; Xue et al.,
2025). As a result, most precipitation from the hills converges into these gully ponds and check dams. A
lower pond water line slope indicates evaporation fractionation during retention. Evaporation
preferentially removes lighter isotopes ($^1$H and $^{16}$O), enriching heavier isotopes ($^2$H and $^{18}$O) and shifting
the pond water's isotopic composition from its precipitation source (Aragu et al., 1998; Zhang and Wu,
2009; Gleeson et al., 2016). This similarity is primarily due to the fact that pond water originates from
rainfall and its runoff in the hilly-gully region (Liu et al., 2011; Ji et al., 2024). Additionally, the isotopic
values of most groundwater in the gully areas are more depleted compared to those of rainfall and pond
water, likely due to the recharge mechanisms and residence times of the different groundwater types
(Ouali et al., 2024). Nevertheless, these values fall within the range of precipitation isotopic values,
leaning towards the more negative end, suggesting that groundwater is likely directly recharged by
significant precipitation events. This is due to the thin unsaturated zone (<10 m) in the gully areas, which
facilitates rapid infiltration and direct recharge from heavy rainfall.
To further investigate the complexity of different types of groundwater in the gully area, we
conducted hydrological and geological surveys, collecting water samples from spring water, pore water,
and fissure water. The results show that the isotopic values of spring water, pore water, and fissure water
are closely clustered, indicating a strong interconnection among the different types of groundwater within
the hydrological cycle. This is likely due to their shared geological and hydrological environments
(Bouwer, 2002; Li et al., 2021; Zhang et al., 2022). Our study found that the water line slope of pore
water and fissure water was higher in the dry season than in the rainy season, with values falling between
the slopes of pond water from the rainy and dry seasons. To investigate the cause of these results, we
analyzed groundwater level dynamics, which showed that water tables were lower at the end of the rainy
season but rebounded afterward, making dry-season tables higher. This suggests that rainy-season
groundwater mixes with evaporatively enriched water, lowering its slope, while dry-season groundwater
is recharged by delayed rainfall and pond water, increasing its slope. These findings reveal that isotopic
composition is influenced by both current and prior hydrological conditions. They also highlight the



complexity of evaporation fractionation regulated by water mixing and demonstrate the significant
impact of pond construction on groundwater recharge and regional hydrology in the gully regions.

**6.2. Groundwater recharge processes in the gully region**

In recent years, discussions of groundwater recharge sources have primarily focused on tableland

and hilly areas with thick loess deposits, although only a few researchers have examined gully regions
(Li et al., 2017; Xiang, 2020; Lu, 2020). For example, Liu et al. (2011) found that groundwater near
valleys in the hilly loess area is replenished by precipitation, runoff, and surface water. Our findings
support this study, identifying rainfall and surface water as major sources of gully groundwater recharge.
We classify groundwater into three types (spring water, pore water, and fissure water) and propose a
gradual recharge process, including (1) direct and indirect recharge from precipitation to pond water and
pore water, respectively; (2) concentrated recharge from pond water to pore water; and (3) percolation
recharge from pore water to fissure water. This reflects the complexity of the groundwater system and
the significant impact of human activities (e.g., ponds and check dams) on hydrological processes.

In the deep-profile unsaturated zones of the hilly region on the Loess Plateau, previous studies have

used chloride mass balance and tritium peak methods to estimate groundwater recharge from
precipitation, typically accounting for 2%–22% of annual rainfall, with water residence times in the
unsaturated zone lasting several years or even hundreds of years (Huang et al., 2011; Tan et al., 2016; Li
et al., 2017; Wang et al., 2024). However, these studies did not address the catchment role of gully regions.
Field observations and past studies show that precipitation rarely directly infiltrates thick loess in hilly
areas (Xu et al., 1993; Li, 2001). Instead, it forms surface runoff that converges into gullies and
accumulates in ponds or other water bodies (perched water), serving as a concentrated recharge source
for groundwater (Yu et al., 2025), reflecting the sustained and delayed impact of gully runoff on
groundwater recharge, which is consistent with the results of this study.

Notably, our study does not consider confined water. Tan et al. (2016) indicated that the groundwater

in the high mountain-hilly loess aquifer does not originate from the upwelling of ancient regional
groundwater, and there is no evidence of deep confined water beneath the loess strata in the high
mountain-hills. Additionally, our findings represent only the groundwater recharge results in the gully
regions for two reasons: 1) The hydrological system is complex, with significant variations across
different landscapes of the Loess Plateau (Li et al., 2019; Li et al., 2021). For example, Li et al. (2019)



found that groundwater dominates the hydrological system in the tableland on the Loess Plateau, where
surface water (streams) is recharged by groundwater because river channels are deeper than the bedrock.
2) Our data collection focused on gully because the "Loess Liang" and "Loess Mao" hillside areas are
covered by thick loess with minimal water sources.

**6.3. Groundwater recharge rates in the gully region**

In many parts of the world, identifying the sources of groundwater recharge and its renewability is

essential for effective water resource management (Ajjur and Baalousha, 2021; Meles et al., 2024). In
the hilly-gully region of the Loess Plateau, where groundwater is considered a crucial source of safe
water, understanding the origins and recharge of aquifers provides valuable information for water
resource planners (Liu et al., 2011; Wang et al., 2024). This knowledge is essential and should be shared
with regions facing similar challenges.

Groundwater recharge can be quantified from three hydrological sources: surface water, the

unsaturated zone, and the saturated zone (Scanlon et al., 2022). Recharge estimates based on the saturated
zone are generally more reliable than those from the unsaturated zone, as the latter represents potential
recharge that may not ultimately reach the groundwater table (Beven and Germann, 2013; Huang et al.,
2019). The groundwater table fluctuation method is widely used for estimating saturated zone recharge
due to its high temporal resolution and intuitive nature (Gumuła-Kawęcka et al., 2022; Xu et al., 2024).
In our study area, ITTPs estimated similar transit times for both pore water and fissure water. Therefore,
we used the groundwater table fluctuation method to assess the recharge of pore water in the gully region.
The total recharge from 2023 to 2024 was estimated at $241.4 \pm 6.0$ mm and $238 \pm 6.0$ mm using the MRC
and RISE methods, respectively. Under constant specific yield conditions, the MRC method typically
estimates higher groundwater recharge and recharge days than RISE, as it accounts for groundwater table
decline due to lateral outflow and other discharge processes in the absence of recharge (Heppner and
Nimmo, 2005). Our findings support this pattern. Furthermore, the key parameter for estimating
groundwater recharge using the groundwater table fluctuation method is specific yield, which depends
on factors such as soil properties and water table depth. This value can be derived from empirical soil
texture, pumping tests, or the test pit method (Nachabe, 2002; Liang et al., 2016). Shah and Ross (2009)
found the test pit method reliable for water tables deeper than 2 m. In this study, with a water table
exceeding 2 m, the specific yield was 0.32, consistent with values of 0.3 in similar soil conditions (Wang



et al., 2023).

Research on groundwater recharge in the Loess Plateau has mainly focused on deep-profile

unsaturated zones in the tableland and hilly areas, with tracer methods estimating recharge between 9 to
100 mm (Huang et al., 2011; Li et al., 2017; Lu, 2020; Wang et al., 2024). In contrast, our study in the
gully region indicates recharge of up to 240 mm, much higher than previous estimates on deep-profile
unsaturated zones. This difference reflects several factors: 1) Unsaturated zone thickness—In the gully
region, the unsaturated zone is generally less than 10 m thick, much shallower than in tableland and hilly
areas (mean thickness of 92.2 m), making infiltration easier and promoting effective recharge. 2) Gully
topography and hydrology—characterized by well-developed channels, concentrated runoff, and
widespread ponds and check dams—promote focused infiltration (Liu et al., 2017; Li et al., 2021; Xue
et al., 2025). 3) Research methods—Tracer methods reflect long-term recharge rates and are better suited
for thicker unsaturated zones (Huang et al., 2011; Lu, 2020; Li et al., 2017). In contrast, the water table
fluctuation method directly captures short-term recharge dynamics and works better in thinner
unsaturated zones. Moreover, this method also better captures surface water-groundwater interactions
and focused recharge effects (Gumuła-Kawęcka et al., 2022). These findings underscore the importance
of studying recharge in gully regions, filling a research gap in the Loess Plateau's geomorphology and
providing new ecohydrological insights. Furthermore, our study demonstrates the potential of artificial
ponds to regulate water resources and enhance recharge, with valuable implications for water
management and ecological engineering.

**6.4. Revised conceptual model**

To convey our evolving understanding of the spatial structure and dynamics in the Gully Region,

we developed a conceptual model that reframes gullies not simply as erosion features but as active
hydrological conduits for groundwater recharge (Fig. 10). This model integrates stable isotope signatures,
variations in chloride concentrations, ITTPs, groundwater level dynamics, and soil physical properties
such as porosity and field water capacity. Through this integrative framework, we elucidate the
transformation of precipitation into various forms of subsurface water by explicitly tracing its movement
through a cascade of compartments—from surface ponding in dammed gullies, to infiltration through the
unsaturated zone, and eventual recharge into both the porous aquifer and underlying bedrock fissure
systems.



This conceptual reframing is grounded in the stark hydrological contrasts between hilly uplands and
gully systems and directly addresses a critical knowledge gap in understanding the hydrological
functioning of gully environments. In the hilly uplands, thick loess deposits—often exceeding 90 m
(including low-permeability aquifers)—combined with steep slopes (>15°) severely restrict vertical
infiltration. Compounded by short-duration, high-intensity rainfall events that provide insufficient
moisture for deep profile wetting, this results in the rapid conversion of rainfall into surface runoff (Li et
al., 2021; Zhu et al., 2018). This runoff is systematically funneled downslope into gully systems, a
process further intensified by engineered interventions such as check dams and retention ponds that
intercept and concentrate overland flow. Because precipitation seldom infiltrates directly into the thick
loess of upland regions (Xu et al., 1993; Li, 2001), most infiltration occurs after surface water
accumulates in gullies—particularly within perched water bodies like ponds—which subsequently serve
as focal points for groundwater recharge (Yu et al., 2025).
Crucially, gully systems possess distinct hydrogeological characteristics: the loess mantle is much
thinner (typically < 25 m), and the soils are dominated by silt loam textures with moderate specific yield
(0.02–0.05) and high field capacity (21–28%). These properties promote transient water storage and
enable temporally delayed, depth-partitioned infiltration. Based on our integrated analyses of stable
isotopes, chloride concentrations, and inverse transit time proxies, we find that gullies function not as
passive erosional features but as active recharge conduits. This conceptualization captures a critical
spatial transition—from runoff generation in the hilly uplands to focused recharge in gully zones—
emphasizing the pivotal role of gully systems in regulating groundwater recharge across the Loess
Plateau landscape.
Combined hydrological monitoring and multi-indicator analysis further reveal that following the
rainy season, infiltration depths on hilly slopes are typically shallow—less than 1 m—while groundwater
levels in gully areas exhibit pronounced rises exceeding 2 m. Recharge estimates based on the water table
fluctuations reach up to 240 mm, far surpassing values observed in deep unsaturated zones of tablelands
and hills (Huang et al., 2011; Li et al., 2017; Lu, 2020; Wang et al., 2024). These results reinforce the
role of gullies as focal points for groundwater recharge and are consistent with prior studies. For instance,
in hilly areas, precipitation rarely recharges groundwater due to the thick loess layers, with an average
infiltration depth of only 1 meter (Wang et al., 2024).
Liu et al. (2011) found that groundwater near valleys in the hilly loess area is replenished by





precipitation, runoff, and surface water. Moreover, fissure water exhibits more depleted isotopic
signatures and higher chloride concentrations, indicating deeper percolation of pore water or mixing with
older recharge sources. These patterns, supported by ITTPs and partial SEM linkages, reveal a
hierarchical recharge sequence: event-driven infiltration enters the shallow pore aquifer, some of which
slowly percolates into deeper fissure zones. This hierarchical mechanism is facilitated by the combination
of thin loess mantles, engineered interventions (e.g., check dams and ponds), and delayed hydrological
responses.
By integrating multiple lines of evidence, this conceptual model redefines gullies as selective
recharge corridors shaped by both geomorphic structure and human intervention. It challenges the
traditional view of gullies as purely erosional landforms and emphasizes their dual hydrological function:
acting both as runoff conveyance channels and as transient reservoirs that store and redistribute water
across space and time. This recharge capacity is jointly governed by topographic convergence, reduced
loess thickness, and the presence of engineered structures such as check dams and retention ponds.
Crucially, the model offers insight into the multifunctionality of ecological engineering—
particularly check dams and ponds—in enhancing hydrological regulation, water security, and ecosystem
restoration across the Loess Plateau. Compared to the traditional piston-flow and preferential flow
models commonly applied in the region, the proposed "gully-dominated preferential recharge mechanism"
marks a notable theoretical advancement. Whereas previous models primarily emphasize vertical
infiltration through homogeneous loess layers, this study is the first to quantitatively identify a cascade
recharge process unique to thin-loess gully catchments. By identifying the pivotal role of gully systems
in stormwater detention, delayed infiltration, and multi-aquifer recharge, this study establishes a robust
theoretical and technical foundation for improving water resource allocation, infrastructure planning, and
groundwater sustainability in arid and semi-arid regions. Beyond advancing theoretical understanding of
regional hydrological processes, it also provides a sound basis for developing spatially targeted models
of groundwater recharge.



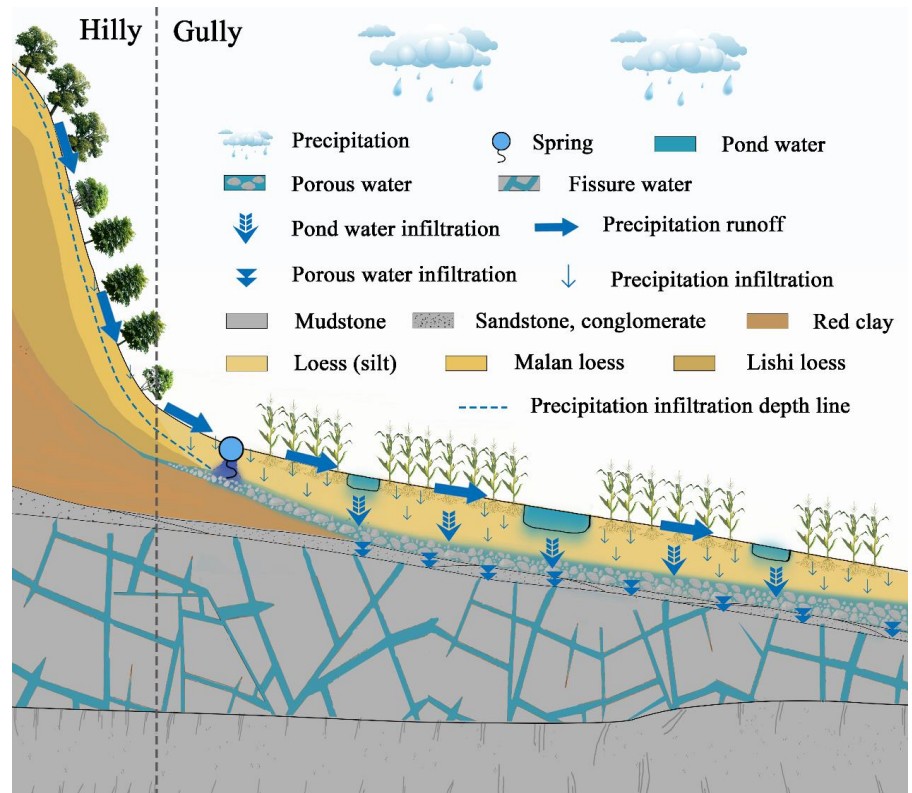

Fig. 10. Hydraulic connections between different water bodies in the hilly-gully region of the Loess

Plateau. The study area consists of hilly and gully regions. In the hilly area, the stratigraphic sequence

from top to bottom is Malan loess, Lishi loess, red clay, sandstone, and mudstone. Rainfall infiltration

within the Malan loess is less than 1 m, and the area is mainly covered by vegetation. In the gully area,

the stratigraphy from top to bottom includes loess (silt), sandstone and conglomerate, and mudstone.

Pore water is found within the sandstone and conglomerate, while fissure water occurs in bedrock

fractures (mudstone). Numerous check dams or ponds are distributed throughout the gully area. The

vertical separation between the pore water and pond water ranges from 3 to 5 m. Corn is the main crop

cultivated in this region. Most springs in the study area are located at the junction of the hilly and gully

regions and are discharged from pore water.

## 6.5. Limitations and future research directions

This study underscores the limitations of relying on a single indicator to infer groundwater recharge

pathways, as doing so may lead to oversimplified or potentially misleading interpretations of complex



hydrological processes. While stable isotope signatures confirm that precipitation contributes to pore
aquifer recharge, they do not provide clear evidence of direct recharge from pond water to either pore or
fissure groundwater during the dry season. In contrast, the spatial distribution of chloride concentrations
offers compelling support for focused pond water leakage into the shallow groundwater system.

Although variance decomposition analysis and structural equation modeling suggest statistical

linkages among different water bodies and provide useful clues about potential recharge pathways, these
associations may be confounded by overlapping isotopic signatures or by non-conservative processes
such as evaporation and mixing, potentially leading to an overestimation of hydrological connectivity. In
the absence of mass balance constraints, the pathways inferred through SEM may not accurately reflect
the actual flow processes within the groundwater system. By contrast, the water table fluctuation method
captures both vertical and lateral recharge processes, yielding estimates that are more likely to reflect
actual recharge rates. Drawing on this empirical evidence, the present study further substantiates the
critical role of gully zones in regional groundwater recharge.

Despite effort to address uncertainties, limitations remain in terms of spatial and temporal sampling

density. For example, the lack of long-term tracers such as groundwater age, combined with limited
observations of groundwater level fluctuations, constrains our ability to assess recharge dynamics over
multi-year timescales. Additionally, the current sampling design includes only two campaigns during the
rainy and dry seasons, which may be insufficient to fully capture the seasonal variability of ITTP values.
This, in turn, may affect the accuracy of groundwater renewal frequency estimates and the strength of
inferred hydrological connections. In arid and semi-arid regions, groundwater recharge is typically
triggered by infrequent, high-intensity rainfall events. However, existing sampling strategies based on
seasonal intervals often lack the temporal resolution necessary to capture these short-lived, event-driven
recharge processes effectively.

Future research should address these issues through several improvements: (1) conducting higher-

frequency, event-scale sampling to systematically monitor rainfall, spring discharge, and pond water
level dynamics, thus capturing the influence of key hydrological events on recharge processes; (2)
expanding the spatial coverage of pore and fissure well monitoring to improve the accuracy of regional
recharge pattern identification; and (3) incorporating additional environmental tracers (e.g., $^{3}H$, $^{22}Na$) to
trace flow paths and estimate recharge lag times. In addition, systematic observation of event-scale
hydrological processes should be strengthened by establishing a high-frequency, event-driven monitoring



network to better capture the nonlinear coupling among rainfall, surface runoff, and groundwater
dynamics. This approach would significantly improve our understanding of rapid infiltration events and
associated recharge mechanisms.
From a methodological perspective, integrating statistical techniques—such as SEM and variance
decomposition analysis—with process-based physical models like MODFLOW and HYDRUS can
provide mechanistically constrained insights into recharge pathways. Compared to correlation-based
statistical methods, physical models offer greater precision in characterizing groundwater flow and
recharge processes across both temporal and spatial dimensions, helping to reduce uncertainties
associated with non-conservative tracer behavior and the absence of mass balance constraints. Regarding
measurements, hydrometric instrumentation within check dams and beneath pond beds could further
quantify the recharge effects of various engineering interventions under specific hydrological conditions.
Additionally, integrating isotopic data with mean transit time modeling, combined with targeted field
monitoring and improved spatial analysis, could help elucidate recharge pathways, quantify temporal
dynamics, and enhance process-level understanding of groundwater recharge in complex dryland
landscapes. Collectively, these efforts will contribute to a stronger theoretical foundation and offer
practical guidance for the precise management of water resources, the design of ecologically appropriate
engineering interventions, and the implementation of effective landscape rehabilitation strategies.

**7. Conclusion**
Through integrated analysis of stable isotopes, chloride concentrations, water table fluctuations, and
inverse transit time proxies, we developed multiple lines of evidence to reframe gullies in the Loess
Plateau as hydrologically significant recharge zones, rather than solely as indicators of erosion and
degradation. Precipitation events drive substantial recharge to shallow pore aquifers, with annual rates
exceeding 238 mm—accounting for a large fraction of annual rainfall. While isotopic evidence for
recharge from pond water is obscured by evaporative fractionation, chloride concentrations provide a
clear signal of subsurface connectivity. Recharge in this system is both spatially concentrated and
temporally selective, shaped by terrain configuration, loess stratigraphy, and ecological engineering
structures such as check dams and ponds.
These findings offer a process-based foundation for developing hydrological indicators of landscape
function and restoration performance in dryland environments. Specifically, recharge magnitude, isotopic



damping, and solute accumulation patterns may serve as diagnostic tools for identifying effective
recharge zones and tracking system responses to intervention. To refine these indicators, future studies
should incorporate high-frequency monitoring, event-based sampling, and multi-tracer approaches.
Collectively, this work challenges conventional views of gullies as hydrological liabilities and
demonstrates their underappreciated role as targeted recharge assets—advancing dryland groundwater
sustainability and providing actionable insights for landscape-scale ecological restoration and
management.



**Author contributions:**
ZXJ: Conceptualization, Methodology, Formal analysis, Investigation, Visualization, Data curation,
Validation, Writing-original draft. ADZ: Conceptualization, Formal analysis, Validation, Visualization,
Writing-review & editing. LW: Conceptualization, Funding acquisition, Project administration,
Resources, Supervision, Visualization, Writing-review & editing.
**Data availability:**
Data are available from the corresponding author upon reasonable request.
**Conflicts of Interest:**
The contact author has declared that none of the authors has any competing interests.
**Acknowledgments:**
This work was supported by the National Natural Science Foundation of China (grant numbers 42171043,
42377318, and U24A20629).







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





 **Appendix A**

1. The typical loess landforms on the Loess Plateau is shown.

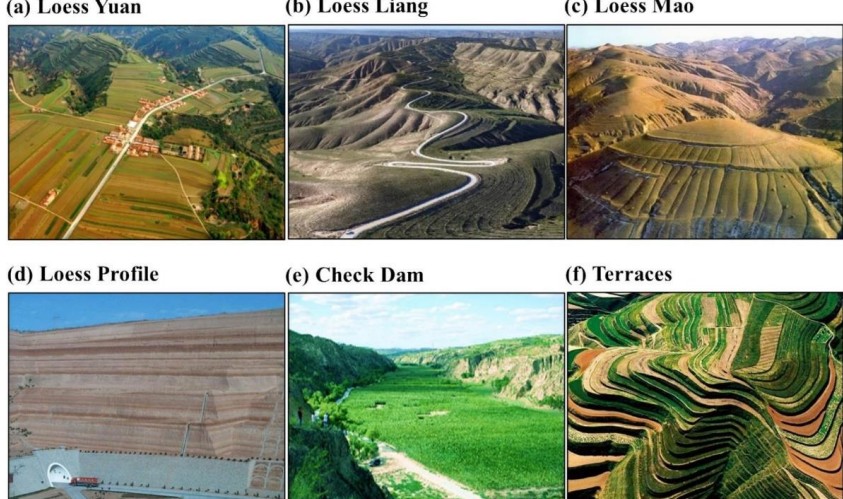


Fig. A1. Typical loess landforms are (a) Loess Yuan, (b) Loess Liang, and (c) Loess Mao. (d) A loess
profile, (e) check dams and (f) terraces on the Chinese Loess Plateau. (adapted from Jia et al., 2024)
2. The specific yield of different soil textures is shown.
Table A1. The specific yield of different texture of soil (adapted from Liang, 2016)

| Texture | Average Specific Yield | Minimum Specific Yield | Maximum Specific Yield | Coefficient of Variation (%) |
|---|---|---|---|---|
| Clay | 0.02 | 0.00 | 0.05 | 59 |
| Silt | 0.08 | 0.03 | 0.19 | 60 |
| Sandy Clay | 0.07 | 0.03 | 0.12 | 44 |
| Fine Sand | 0.21 | 0.10 | 0.28 | 32 |
| Medium Sand | 0.26 | 0.15 | 0.32 | 18 |
| Coarse Sand | 0.27 | 0.20 | 0.35 | 18 |
| Gravelly Sand | 0.25 | 0.20 | 0.35 | 21 |
| Fine Gravel | 0.25 | 0.21 | 0.35 | 18 |
| Medium Gravel | 0.23 | 0.13 | 0.26 | 14 |
| Coarse Gravel | 0.22 | 0.12 | 0.26 | 20 |



3. The isotopic composition (δ²H and δ¹⁸O) of various water sources in the rainy and dry seasons is shown.
Table A2. Isotopic composition (δ²H and δ¹⁸O) of various water sources in the rainy and dry seasons

|  | Rainy season | | Dry season | |
|---|---|---|---|---|
|  | δ²H | δ¹⁸O | δ²H | δ¹⁸O |
| Rainfall | −36.6±20.4‰ | −5.6±2.3‰ | −31.0±23.2‰ | −4.9±3.0‰ |
| Pond water | −40.5±13.1‰ | −4.1±2.3‰ | −24.5±6.9‰ | −0.8±1.3‰ |
| Spring water | −67.3±2.6‰ | −9.0±0.4‰ | −68.4±2.2‰ | −9.0±0.4‰ |
| Pore water | −66.3±3.1‰ | −9.0±0.6‰ | −65.4±3.8‰ | −8.5±0.6‰ |
| Fissure water | −65.0±3.8‰ | −8.8±0.9‰ | −64.5±5.5‰ | −8.5±0.9‰ |

4. The chloride concentrations of various water sources in the rainy and dry seasons is shown.

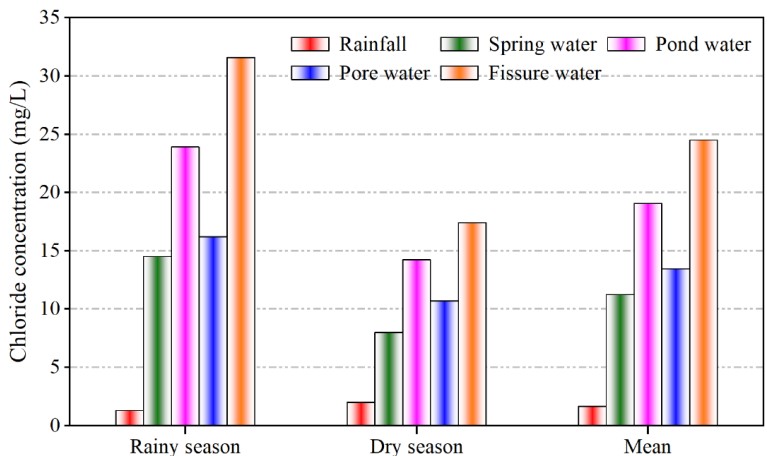


Fig. A2. Chloride concentrations of various water sources in the rainy and dry seasons.