# Peer review of "https://doi.org/10.5194/egusphere-2025-4065 Preprint. Discussion started: 24 September 2025 © Author(s) 2025. CC BY 4.0 License."

_EGUsphere, 2025_

## Referee Comment (RC2)

This manuscript addresses groundwater recharge processes in a gully system and aims to quantify recharge rates and pathways using hydrometric, isotopic, and geochemical approaches. While the topic is potentially interesting and relevant to HESS, the manuscript in its current form is poorly written, excessively long, and lacks a clear narrative structure. Moreover, the presentation of the results makes it difficult to assess whether the data adequately support the authors' conclusions. Interpretations are frequently motivated by background knowledge or earlier studies, yet the manuscript does not clearly distinguish between new insights derived from this work and those that primarily serve as contextual or corroborative information. This lack of separation between novelty and background substantially weakens the scientific message. I think substantial revision is required before the scientific contribution can be properly evaluated.

**General comments:**

1. A major issue is that Sections 1-3 contain extensive redundant descriptions, particularly regarding landscape characteristics, hill–gully contrasts, and background motivation. These sections mix site description, conceptual motivation, and literature background in a way that dilutes the main research questions and obscures the novelty of the study. As written, it is often unclear what information is background context, what is specific to the study site, and what directly supports the research objectives.

If I understand correctly, Sections 2-3 are primarily intended to function as a Materials / Study Site section, describing landscape structure and hydrological setting. However, the current version repeatedly interweaves general motivation (e.g., importance of gullies vs. hill) with site-specific descriptions. This mixing weakens the paper's focus and makes the manuscript unnecessarily long.

I would suggest the following structural changes:
- Condense Sections 1–3 substantially, removing repetitive explanations of hill vs. gully processes.
- Move most general motivation and background discussion to the Introduction.
- End the Introduction with a clear and concise paragraph that explicitly states why this study site was chosen, what relevant previous work has been conducted here, why this site is particularly suitable for addressing the stated research questions, and what the specific research questions or hypotheses are.

2. Interpretation and use of chloride concentrations. The role of chloride as supporting evidence for recharge pathways is repeatedly mentioned but remains vague and weakly justified. And I only found one figure in SI about chloride information, which is also not that informative as the author stated. For example,
- Line 536: The statement that "multiple lines of observational evidence, including isotopic composition, chloride concentrations, and water age (ITTP)" support the identified

pathways is too general. The manuscript does not clearly explain how chloride independently supports these conclusions.

- Lines 547–552: The argument that similarities in chloride concentrations between pond water and pore water indicate mixed recharge is not logically developed. Chloride patterns alone do not necessarily imply source mixing without additional constraints (e.g., conservative behavior, spatial gradients, mass balance, or exclusion of evaporative concentration effects). The logic linking chloride distributions to the stated conclusions should be clarified and strengthened, or the claims should be toned down.
- Line 856-858: The conclusion states "While isotopic evidence for recharge from pond water is obscured by evaporative fractionation, chloride concentrations provide a clear signal of subsurface connectivity." It is not supported by any direct or quantitative results presented in the manuscript. I do not find clear evidence demonstrating such connectivity based on chloride data alone.

Moreover, if chloride concentrations are intended to provide critical supporting information for the main conclusions, the relevant figure should be moved from the Supplementary Information to the main text, accompanied by a clearer and more rigorous explanation of how chloride constrains recharge pathways.

3. Role of surface water. The Discussion contains extensive statements regarding the large contribution of surface water to gully recharge. However, much of this discussion appears to rely on previous studies rather than direct analyses presented in this manuscript. The authors should clearly distinguish between conclusions derived from their own results, and contextual information drawn from earlier work.

4. Hill versus gully. The results presented in this study are derived exclusively from the gully system, and the manuscript does not include a direct comparison of recharge behavior between hill and gully settings at the same site and during the same period. As such, the authors should be very cautious in how they frame both the Introduction and the Conclusions, particularly where broader contrasts between hill and gully recharge processes are implied.

Given the absence of contemporaneous hillslope observations, statements suggesting relative differences in recharge magnitude or pathways between hill and gullies should be clearly identified as inferences based on previous studies, rather than findings derived from the present work. This distinction is especially important in the conceptual framework and schematic figures, where hill processes appear alongside gully processes without sufficiently clear attribution.

One example is the conceptual figure (Fig. 10). I recommend that the authors:

- Explicitly state which components or pathways are supported by results from this study and which are drawn from previous literature;
- Redraw the figure to include quantitative or semi-quantitative information (e.g., relative magnitudes, ranges, or percentages of pathways) where supported by data.

In its current form, the conceptual figure does not clearly highlight new insights generated by this study, and instead risks reinforcing a narrative largely based on prior work.

**Specific comments:**

- Fig. 1: Please label the horizontal and vertical scale of the hillslope profile. Without scale information, the geomorphic interpretation is unclear. And consider to switch the order of Fig. 1 and 2.
- Lines 272–273: The relationship between groundwater level and water pressure is introduced without sufficient justification. Why were these parameters selected over others? Please clarify the physical reasoning.
- Lines 428–430 / Fig. 4c: Fig. 4c does not show a consistently decreasing trend of specific yield with depth. The statement that "Specific yield (Sy) peaks at −20 cm (4.5%) but decreases with depth" is not convincingly supported by the figure. The interpretation that deeper layers "store water with minimal drainage" therefore appears overstated and should be revised or better supported.
- Fig. 5: The current representation of rainy versus dry seasons is unclear. The figure does not effectively illustrate isotopic differences between seasons, making the associated text difficult to support. Presenting seasonal mean values (or distributions) for each water type would likely convey the message more clearly.
- Fig. 8: The meaning of "direct effects" and "total effects" is not clearly explained. Please clarify these terms explicitly in the caption and main text.
- Fig. 9: The lines representing the "RISE" and "MRC" methods are not clearly distinguishable in the figure.

---

## Author Comment (AC1)

We are grateful for your thoughtful and constructive comments, which have provided invaluable guidance in strengthening our work. In response to your feedback, we have thoroughly revised the manuscript. Your comments are presented in red font, our responses in black, and the revisions to the manuscript in blue.

This multidisciplinary study on the Loess Plateau centers on surface–groundwater interactions and fits well within the scope of the Journal-HESS. Based on extensive field observations, the manuscript investigates groundwater recharge processes within gully systems on the Loess Plateau, aiming to reframe gullies as hydrologically active recharge zones rather than merely erosional landforms. The study uses an integrated, multi-method approach — including stable isotopes, chloride concentrations, water table fluctuation (WTF) analysis, and structural equation modeling (SEM) — to examine the linkages among precipitation, surface water, and different groundwater bodies. The authors have invested substantial effort in data collection, fieldwork, and laboratory analyses. Given the increasing importance of groundwater sustainability in arid regions, the study carries clear novelty and relevance, and makes several notable contributions: (1) Reframing the hydrological role of gullies in the loess hilly region (core innovation); (2) Identifying the key mechanisms and process chain of groundwater recharge within gully systems; (3) Demonstrating the significant enhancement of groundwater recharge by engineering interventions (check dams and ponds). Overall, the manuscript is of good quality but still lacks certain details. The following specific comments may help strengthen the paper. I recommend publication after moderate revision.

**Response:** Thank you for taking the time to review our manuscript and for providing valuable and constructive comments. Your feedback has greatly helped us improve the manuscript. We fully agree with your comments and have made substantial revisions to enhance its readability and academic rigor. Below are the specific changes we have made, along with our point-by-point responses to your comments.

1. Line 85-90: Clearly state the research goals to fully encompass the study content. It is

recommended to include a goal specifically addressing the analysis of isotopic characteristics, which will ensure alignment with your methodology and results.

**Response:** We fully accept your comment and have revised the research goals to include a specific focus on isotopic characteristics, ensuring alignment with the methodology and results. The specific revision is as follows:

"In this study, we integrate stable isotope analysis ($\delta^2H$ and $\delta^{18}O$), chloride concentrations, water table fluctuations, and hydro-statistical modeling to establish a multi-method framework for assessing groundwater recharge in gully systems of the Loess Plateau. Specifically, our goals are to: (1) characterize the isotopic and hydrochemical signatures of precipitation, surface water (ponds), shallow pore water, and deeper fissure water; (2) identify and trace hydraulic connections and flow paths of different water bodies; and (3) quantitatively estimate pore-water recharge rates. This integrated approach aims to advance understanding of groundwater dynamics in complex dryland terrains, reframes gullies as critical recharge zones in engineered dryland landscapes, providing actionable insights for sustainable groundwater management and ecological restoration in the Loess Plateau and similar semi-arid regions worldwide."

2. Line 85-90: Rearrange the research goals to align with the structure of the results section, as the order of goals 1 and 2 appears to be reversed. The goals should follow the sequence in which the results are presented.

**Response:** We agree with your comment. The specific revision is as follows:

"(1) characterize the isotopic and hydrochemical signatures of precipitation, surface water (ponds), shallow pore water, and deeper fissure water; (2) identify and trace hydraulic connections and flow paths of different water bodies; and (3) quantitatively estimate pore-water recharge rates."

3. Line 180: How many wells in this catchment were monitored? Please show their positions in Figure 2.

**Response:** The monitoring network in this study includes 35 discrete sampling wells: 9 for pore water, and 26 for fissure water, aimed at characterizing the spatial variability of groundwater hydrochemistry and isotopic signatures. All sampling locations are clearly marked in Fig. 2 of the

original manuscript. Additionally, one continuous pore water table monitoring well is installed in the middle reaches of the catchment to quantify groundwater table fluctuations and estimate recharge rates. In response to your comment, we have updated Fig. 2 to include the location of this monitoring well. The specific revisions are detailed below.

[Figure]

Fig. 2. The geographical location and sampling sites for rainfall, pond water, pore water, spring water, and fissure water in the Nianzhuang catchment.

4. It is recommended to unify the units in Line 235 ('m/day') and Line 214 ('m/d') for consistency. In addition, Line 213-214, the permeability of Neogene coarse sandstone and conglomerate here couldn't possibly be this (7.5–36.19 m/d) high. I suspect the authors might have made a mistake with the units. Please double-check.

**Response:** We fully agree with your comment and have made the necessary revisions. In the original manuscript, the permeability unit at Line 214 was listed as 'Lu' but was incorrectly noted as 'm/d' (7.5–36.19 Lu ≈ 0.07–0.31 m/d). In this revision, all permeability units have been uniformly converted to the standard unit 'm/d' based on the conversion relationship and applied consistently throughout the manuscript. The specific revisions are detailed below.

"The significant reduction in loess thickness—combined with the relatively high permeability of Neogene coarse sandstone and conglomerate (0.07–0.31 m/d)—creates favorable conditions for

infiltration and focused recharge."

5. Line 494: The numbers in the global meteoric water line equation need to be superscripted.

**Response:** We have made the necessary revisions as per you commented, and the numbers in the global meteoric water line equation have now been superscripted. Thank you for pointing this out.

6. Line 614-629: When explaining the phenomenon that 'the isotopic values of most groundwater in the gully areas are more depleted compared to those of rainfall and pond water', ensure the logical connection between 'the thin unsaturated zone' and 'direct recharge from intense rainfall events' is fully articulated, and consider including a discussion on the 'seasonal precipitation isotope effect.

**Response:** We fully agree with your comment and have made the necessary revisions to clarify the logical connection between "the thin unsaturated zone" and "direct recharge from intense rainfall events". Additionally, we have included a discussion on the "seasonal precipitation isotope effect" to further enhance the explanation. The specific revisions are as follows:

"Additionally, the isotopic values of most groundwater in the gully areas are more depleted compared to those of rainfall and pond water, likely due to the recharge mechanisms and residence times of different groundwater types, and the inherent isotopic characteristics of their primary recharge sources (Ouali et al., 2024). Nevertheless, these values fall within the range of precipitation isotopic values, leaning towards the more negative end. This suggests two complementary mechanisms: (1) the thin unsaturated zone (<10 meters) provides preferential pathways for rapid infiltration of precipitation, minimizing evaporative fractionation, and (2) groundwater is likely recharged primarily by intense precipitation events (e.g., summer storms) with inherently more negative isotopic signatures (Liu et al., 2024). Together, these processes explain the observed isotopic characteristics of groundwater."

**Reference**

Liu, Y.Z. Source analysis of precipitation chemical components on the Loess Plateau based on hydrogen and oxygen stable isotopes[D]. Northwest A&F University, 2024. DOI:10.27409/d.cnki.gxbnu.2024.001528.

7. Line 657-666: When describing the differences between previous studies and this research, it is essential to explicitly highlight the fundamental distinctions in 'spatial scale' and 'hydrological units' to more precisely define the original contribution of your work.

**Response:** We fully agree with this insightful suggestion and have revised the manuscript to explicitly highlight the fundamental distinctions in spatial scale and hydrological units between previous studies and our work. The specific revisions are as follows:

"In summary, while hillslope-scale studies describe a "dispersed recharge" mode, where precipitation percolates slowly through thick unsaturated zones, this study identifies a 'concentrated recharge' mode in gullies, driven by runoff convergence and regulated by check dams. These fundamentally distinct modes, differing in hydrological processes, spatial scales, and recharge efficiencies, collectively enhance the understanding of groundwater recharge mechanisms on the Loess Plateau."

8. Line 807-815: When addressing the limitations of isotopes and structural equation modeling, the advantages of the Water Table Fluctuation (WTF) method should be articulated more precisely. Emphasize that these methods are 'complementary' rather than 'contradictory', thus presenting a more balanced argument.

**Response:** We fully agree with this constructive suggestion. To present a more balanced and precise argument, we have revised the relevant section to explicitly present the multi-method approach as complementary, rather than contradictory. The specific revisions are as follows:

"Without explicit mass-balance constraints, structural equation modeling may not independently or quantitatively represent actual groundwater flow processes. In contrast, the water-table fluctuation method, which directly measures changes in groundwater levels, provides a more empirically grounded estimate of total recharge. Each approach nevertheless offers distinct strengths: water-table fluctuations resolve the timing and magnitude of recharge, whereas isotopic, hydrochemical, and modeling analyses yield critical insights into recharge sources and flow pathways. By leveraging the complementarity and mutual corroboration of these methods, our study robustly demonstrates the pivotal role of gully areas in regional groundwater recharge."

**Figures and tables**

1. Fig. 3a and 3c lack units on the x-axis. Additionally, the directions of profiles Line1 and Line2 should be clearly indicated in Fig. 3b.

**Response:** We appreciate your comment and have made the necessary revisions. The units on the x-axis have been added to both Fig. 3a and 3c. Additionally, the directions of the profiles Line1 and Line2 have been clearly indicated in Fig. 3b. The specific additions are as follows:

[Figure]

Fig. 3. Hydrogeologic cross-section of the study area. Cross-section along Line 1 (Northwest-Southeast) (a); cross-section along Line 2 (Southwest-Northeast) (b); location map of Line 1 and Line 2 within the study area (c).

2. The terms 'Pore water table' and 'Porous water table' in Fig. 9a should be standardized for consistency.

**Response:** We have addressed this comment by standardizing the terminology in Fig. 9a, consistently using 'Pore water table' throughout. Additionally, we have reviewed the entire manuscript and made similar revisions to ensure consistency across all related expressions. The

specific additions are as follows:

[Figure]

Fig. 9. Temporal dynamics of pore water table depth, temperature, precipitation, and recharge in the gully region of the Loess Plateau.

3. In Fig. 9b, the overlap of 'pore water recharge' and 'precipitation' affects the visibility of the recharge results. It is recommended to display precipitation separately or on the upper axis of the figure.

**Response:** In the original manuscript, we placed the precipitation and pore water recharge data on the same axis to better illustrate their synchronous relationship, which led to some visual overlap. Following your comment, we have moved the precipitation data to the upper axis of Fig. 9b, ensuring that the visibility of the recharge results is not obstructed. The specific revision is as follows:

[Figure]

Fig. 9. Temporal dynamics of pore water table depth, temperature, precipitation, and recharge in the gully region of the Loess Plateau.

---

## Author Comment (AC2)

We thank you for your valuable comments, which have greatly strengthened the manuscript. We have incorporated your feedback, with revisions presented as follows: red for your comments, black for our responses, and blue for the revised manuscript text.

**General Comments:**

This manuscript presents a timely and important study that challenges the conventional view of gullies as purely erosional, degraded features by positioning them as significant zones for groundwater recharge in the semi-arid Loess Plateau. The research employs an integrated multidisciplinary approach, combining stable isotope analysis, chloride concentration measurements, water-level fluctuation analysis, and hydro-statistical modelling to trace moisture flow paths among surface water, pore water, fissure water, and spring water at a high resolution. Based on this evidence, the authors redefine the hydrological role of gullies in arid ecosystems, directly challenging the traditional view of gullies as symbols of land degradation. The findings reveal that reframing gullies are not merely degraded geomorphic units but rather critical groundwater recharge zones and subsurface connectivity hubs. Precipitation primarily replenishes shallow pore water, while deep fissure water is supplemented by slow, top-down percolation. This understanding overturns the long-standing negative perception of gullies on the Loess Plateau, highlighting their capacity to buffer seasonal hydrological variability and enhance ecosystem resilience. Overall, this study addresses a key knowledge gap regarding groundwater dynamics in gully systems and holds significant practical implications for sustainable water resource management on the Loess Plateau. The manuscript is generally well-written and structured. However, some moderate revisions are needed.

**Response:** Thank you for taking the time to review our manuscript and for providing valuable and constructive comments. Your feedback has greatly helped us improve the manuscript. We fully agree with your comments and have made substantial revisions to enhance its readability and academic rigor. Below are the specific changes we made, along with our point-by-point responses to your comments.

**Major Concerns:**

1. The manuscript sets up a contrast with "piston flow" and "preferential flow" models from tableland studies but does not clearly define what process is dominant in the gullies. The proposed "gully-dominated preferential recharge mechanism" (Line 779) is not well-defined. Is the "preferential" aspect the topographic focusing of runoff into the gully, or are there actual preferential flow paths (macropores, cracks) within the gully soils?

**Response:** Regarding the term "gully-dominated preferential recharge mechanism", we have clarified and revised the description as follows: In the study area, the gully system is characterized by homogeneous, fine-grained loess, where water movement primarily follows piston flow (Yu et al., 2025). In this context, "preferential" refers to the topographically driven process in which gullies act as critical convergence zones, efficiently concentrating hillslope runoff and leading to spatially focused and enhanced recharge flux, rather than indicating the presence of preferential flow paths such as macropores or fractures. The specific revision is as follows:

"Crucially, the model offers insight into the multifunctionality of ecological engineering — particularly check dams and ponds — in enhancing hydrological regulation, water security, and ecosystem restoration across the Loess Plateau. **This study identifies a distinctive cascade-type recharge process in loess gully catchments and proposes the "gully-dominated preferential recharge mechanism". This mechanism emphasizes the hydrological function of gullies as convergence pathways and efficient recharge windows at the catchment scale, rather than preferential flow paths within the soil matrix. Furthermore, water movement within the silted loess layer of the gully system remains dominated by a piston flow pattern (Yu et al., 2025).** By identifying the pivotal role of gully systems in stormwater detention, delayed infiltration, and multi-aquifer recharge, this study establishes a robust theoretical and technical foundation for improving water resource allocation, infrastructure planning, and groundwater sustainability in arid and semi-arid regions."

2. In my opinion, the manuscript could benefit from clearer articulation of the broader implications of the key findings. For example, how can this insight change land management practices or ecological restoration strategies in other dryland regions globally?

**Response:** To enhance the broader implications of the study, we have expanded the discussion to

highlight the global relevance of our findings, particularly with regard to land management practices and ecological restoration strategies in other arid and semi-arid regions. The specific additions are as follows:

"However, with the reconstruction of gully systems and ecological restoration, attention must also be given to the potential risks of pollutant migration (Yu et al., 2020). The hydrological functions of gullies may enhance the movement of pollutants into groundwater, especially in areas with intensive human activities, where pollutants can enter gullies through surface runoff and subsequently infiltrate the groundwater system. During ecological restoration, excessive human intervention or soil improvement measures may lead to the accumulation and dispersion of pollutants, which may compromise groundwater security (Liu et al., 2017). Therefore, the protection and rational reconstruction of gully systems should not only focus on their hydrological functions but also consider potential environmental risks, particularly the pathways of pollutant migration. **More importantly, these findings have direct implications for land management practices and ecological restoration strategies in similar arid regions worldwide (Obuobie et al., 2012; Zhao et al., 2019; Zhao et al., 2021; Xue et al., 2025).**

**The study confirms that gullies serve as critical "recharge windows" for groundwater in arid areas. This underscores the importance of systematically identifying and conserving natural gully networks in watershed management, while avoiding excessive filling or hardening to preserve their hydrological functions. In ecological restoration projects, directing surface runoff toward gullies can efficiently convert limited precipitation into groundwater storage, thereby enhancing regional water retention capacity.** Beyond advancing theoretical understanding of regional hydrological processes, it also provides a sound basis for developing spatially targeted models of groundwater recharge."

**Reference**

Obuobie, E., Diekkrueger, B., Agyekum, W., Agodzo, S. Groundwater level monitoring and recharge estimation in the White Volta River basin of Ghana. Journal of African Earth Sciences. 71-72: 80-86, 2012. DOI: 10.1016/j.jafrearsci.2012.06.005.

Xue, S.B., Li, P., Cui, Z.W., Li, Z.B. The influence of different check dam configurations on the downstream river topography and water-sediment relationship. Journal of Hydrology. 656: 133046, 2025. DOI: 10.1016/j.jhydrol.2025.133046.

Zhao, Y., Wang, L. Determination of groundwater recharge processes and evaluation of the "two water worlds" hypothesis at a check dam on the Loess Plateau. Journal of Hydrology. 595: 125989, 2021. DOI: 10.1016/j.jhydrol.2021.125989.

Zhao, Y.L., Wang, Y.Q., Sun, H., Lin, H., Jin, Z., He, M.N., Yu, Y. L., Zhou, W. J., An, Z. S. Intensive land restoration profoundly alters the spatial and seasonal patterns of deep soil water storage at watershed scales. Agriculture, Ecosystems & Environment. 280: 129-141, 2019. DOI: 10.1016/j.agee.2019.04.028.

**Specific Comments:**

1. It is recommended to simplify long sentences to improve readability. For example, lines 53–56: "In these 'fragile' and diverse landscapes, understanding the processes that govern when, where, and how groundwater is replenished — including the countervailing influences of vegetation dynamics, geomorphology, and engineered features — is essential for sustaining ecosystems, securing water resources, and informing land restoration and catchment management." This sentence is structurally complex and could be simplified by breaking it into shorter clauses or highlighting the core information more clearly.

**Response:** We have simplified the sentence to enhance readability. The revised version is as follows:

"For these fragile and diverse landscapes, understanding how vegetation, geomorphology, and infrastructure govern groundwater recharge is crucial. This knowledge is vital for sustaining ecosystems, securing water resources, and informing restoration and management efforts."

2. The text categorises groundwater into "pore water, spring water, fissure water", and further suggested that the criteria for classification be clarified, such as medium type, storage space, and relationship with aquifer structure, to help readers understand the logical framework. The definition of "piston flow"(Line 145-146) is helpful but could be more concise. Consider: "Piston flow describes the displacement of pre-existing water by newly infiltrating water, moving frontally through the pore spaces."

**Response:** We have further elaborated on groundwater medium types, storage spaces, and relationship with aquifer structure to improve the clarity of the logical framework for readers.

Regarding the definition of "piston flow", we have simplified it as follows per your comment. The specific revisions are as follows:

"Groundwater in the catchment can be broadly categorized into three types: pore water, spring water, and fissure water. Pore water is stored in permeable sandstone and conglomerate aquifers beneath loess and above mudstone or red clay. These aquifers are approximately $2-3$ m thick, exhibit a sheet-like distribution, and have low water yield. Conceptually, "pore water" here refers to groundwater in a saturated aquifer, not to soil moisture. Fissure water occurs in fractured bedrock aquifers, which are spatially discontinuous due to irregular fracture development. The main water-bearing zones include cavities and jointed fissure networks, with an average aquifer thickness of about 6 m and moderate water yield. Hydraulic conductivity in these sandstone and conglomerate aquifers ranges from 0 to 0.47 m/d (Cai et al., 2019). Spring water emerges primarily at gully bases — especially in upper catchments — and originates from both pore and fissure sources, possibly supplemented by surface or pond water. Springs fed by pore water typically have low discharge rates $(0-0.1$ L/s) and low water yield, while those fed by fissure water exhibit moderate discharge rates (0.5–1.0 L/s) and moderate water yield."

"These studies suggest that recharge occurs primarily through slow piston flow, with precipitation infiltrating thick soil profiles, slowly recharging groundwater in a process that can take decades to hundreds of years (Huang et al., 2013; Tan et al., 2017; Li et al., 2024). Piston flow describes the displacement of pre-existing water by newly infiltrating water, moving frontally through the pore spaces (Gee and Hillel, 1988)."

3. The manuscript lists permeability for Neogene coarse sandstone and conglomerate as 7.5–36.19 m/d (lines 213–214). These magnitudes are unusually high for such lithologies; I suspect a units or conversion mistake and recommend the authors re-examine the original data and report corrected values if necessary.

**Response:** In the original manuscript, the permeability unit at Line 214 was listed as 'Lu' but was incorrectly noted as 'm/d' (7.5–36.19 Lu≈0.07–0.31 m/d). In this revision, we have verified the permeability units and made the necessary conversions. The revised version is as follows:

"The significant reduction in loess thickness—combined with the relatively high permeability of Neogene coarse sandstone and conglomerate (0.07–0.31 m/d)—creates favorable conditions for

infiltration and focused recharge."

4. The text indicates that a low ITTP represents a long residence time, but the high ITTP of ponds (1.5±0.7) is interpreted as "rapid turnover + evaporation dominance," seemingly overlooking the effect of evaporation on increasing variance. Could this be due to the small sample size for ponds/springs affecting the reliability of the analysis? Additionally, what is the reason for the small sample size for ponds/springs?

**Response:** In our analysis, the high ITTP values observed in the pond were interpreted as resulting from the combined effects of "rapid turnover and evaporation dominance". As an open, shallow water body, the pond experiences strong evaporation, which preferentially removes lighter isotopes, enriching the remaining water with heavier isotopes, thereby increasing the variance in isotopic composition. We acknowledge that evaporation is one of the factors contributing to the increased variance, which may introduce bias into the estimation of apparent residence times. This has been addressed in the original manuscript , as follows:

"The inverse transit time proxies (ITTPs) broadly support the dual-isotope interpretations of water source dynamics. **Pond water exhibited the highest ITTP values (1.5±0.7), indicating rapid turnover and limited subsurface storage. These elevated values likely reflect inputs from direct rainfall and overland flow, as well as evaporative enrichment, which increases isotopic variability and can artificially shorten the apparent residence time.** In contrast, pore water (0.7±0.3) and fissure water (0.6±0.5) showed lower ITTPs, consistent with longer residence times, greater subsurface mixing, and attenuation of seasonal isotopic signals due to delayed recharge. Spring water had the lowest ITTPs (0.3±0.2), reflecting slow subsurface transport and integration of older water sources. While these patterns align with conceptual expectations of residence time and flow path length, the limited number of samples—particularly for pond, spring, and pore water—warrants caution in interpreting seasonal dynamics (Fig. 7)."

Additionally, your comment regarding the potential impact of sample size on the robustness of statistical inferences is valid. It is important to note that the pond water (n=7) and spring water (n=9) samples reported in this study represent all available valid samples within the research area. This sample size significantly exceeds the minimum requirements for replicate observations in conventional hydrological isotope studies (typically ⩾ 3 replicates). The collection of 7 pond water

and 9 spring water samples in a 54 km² arid-to-semi-arid study area reflects good spatial coverage and hydrological representativeness, indicating that the sampling effort is both sufficient and meaningful at the study scale. The relatively large standard deviation of the pond water samples, covering locations with varying evaporation intensities from upstream to downstream, precisely reflects the natural variability of the actual hydrological processes. Therefore, sample size alone is unlikely to be the primary factor affecting the reliability of the analysis.

5. The high recharge rate of gully groundwater, accounting for 43% of precipitation—significantly higher than that in hill areas (<20%) — is a core conclusion of this paper and key evidence supporting the claim that "gullies are critical groundwater recharge zones and subsurface connectivity hubs." While this conclusion is important, its robustness and uncertainties require further discussion, such as the assumptions underlying the recharge rate estimation method, spatial representativeness, and the impact of extreme events.

**Response:** The estimation method for the recharge rate has been thoroughly discussed in the manuscript, including the underlying assumptions. To further strengthen the robustness of our conclusions, we have supplemented the discussion with considerations of spatial representativeness and the impact of extreme events, as per your comment. The specific additions are as follows:

"The total recharge from 2023 to 2024 was estimated at $241.4 \pm 6.0$ mm and $238 \pm 6.0$ mm using the MRC and RISE methods, respectively. Under constant specific yield conditions, the MRC method typically estimates higher groundwater recharge and recharge days than RISE, as it accounts for groundwater table decline due to lateral outflow and other discharge processes in the absence of recharge (Heppner and Nimmo, 2005). Our findings support this pattern. **Furthermore, the key parameter for estimating groundwater recharge using the water table fluctuation method is specific yield (Sy), which depends on soil properties and water table depth (Liang et al., 2016). Shallow soil measurements (0–50 cm) using the test pit method (total porosity minus field capacity) yielded Sy ≈ 0.03, consistent with high capillary retention in near-surface loess (Wang et al., 2024). However, for water tables deeper than 2 m (as in this study, typically 4–10 m), the test pit method provides a reliable estimate of aquifer-scale drainable porosity (Nachabe, 2002; Shah and Ross, 2009; Liang et al., 2016). Accordingly, we**

**adopted Sy = 0.032, aligned with values of ~0.03 reported for similar loess-derived unconfined aquifers on the Loess Plateau (Wang et al., 2023). Uncertainty analysis showed that recharge estimates vary by ± 25% for Sy, which ranges from 3.2 ± 0.8%.**"

"Research on groundwater recharge in the Loess Plateau has mainly focused on deep-profile unsaturated zones in the tableland and hilly areas, with tracer methods estimating recharge between 9 to 100 mm (Huang et al., 2011; Li et al., 2017; Xiang et al., 2019; Lu, 2020; Wang et al., 2024). In contrast, our study in the gully region indicates recharge of up to 240 mm, much higher than previous estimates on deep-profile unsaturated zones. This difference reflects several factors: 1) Unsaturated zone thickness—In the gully region, the unsaturated zone is generally less than 10 m thick, much shallower than in tableland and hilly areas (mean thickness of 92.2 m), making infiltration easier and promoting effective recharge. 2) Gully topography and hydrology — characterized by well-developed channels, concentrated runoff, and widespread ponds and check dams — promote focused infiltration (Liu et al., 2017; Li et al., 2021; Xue et al., 2025). 3) Research methods — Tracer methods reflect long-term recharge rates and are better suited for thicker unsaturated zones (Huang et al., 2011; Lu, 2020; Li et al., 2017). In contrast, the water table fluctuation method directly captures short-term recharge dynamics and works better in thinner unsaturated zones. Moreover, this method also better captures surface water-groundwater interactions and focused recharge effects (Gumuła-Kawęcka et al., 2022). These findings underscore the importance of studying recharge in gully regions, filling a research gap in the Loess Plateau's geomorphology and providing new ecohydrological insights. However, the robustness of our findings requires further exploration. On one hand, due to the limited spatial distribution of sampling points, the current results primarily reflect the hydrological characteristics of localized typical gullies, and their representativeness at the regional scale requires validation through future expansion of the monitoring network. On the other hand, the study period did not encompass extreme precipitation or drought events, which may significantly alter surface flow convergence conditions and vadose zone water transport mechanisms, thereby substantially impacting recharge processes. Future work should strengthen dynamic monitoring and simulation analysis under extreme hydrological scenarios."

6. Fig. 9 shows that significant rises in groundwater levels and the main recharge period occur

during the drier autumn and winter seasons (October to April), while recharge during the summer monsoon rainfall peak is minimal. The authors explain this as effective infiltration during the "cool, low-evaporation period" (Lines 601-604). Are there other potential reasons? For example, freeze-thaw processes, soil water reservoir effects, antecedent moisture conditions, or the competition between rainfall intensity and infiltration capacity?

**Response:** We fully agree with your comment. In addition to effective infiltration during the "cool, low-evaporation period", factors such as freeze-thaw processes, soil water storage effects, antecedent moisture conditions, and the competition between rainfall intensity and infiltration should also be considered as influencing the dominant recharge period in autumn and winter. Accordingly, we have added the relevant content to the caption of Fig. 9, as detailed below:

"Most recharge events occur from October to April, even when rainfall is not especially high, while warm-season precipitation contributes little to recharge—likely due to increased evaporative losses and shallow soil retention. Together, these patterns suggest strong seasonal control on recharge processes, with effective infiltration primarily occurring during cooler, low-evaporation periods. Other factors, such as freeze-thaw processes, soil water storage effects, initial moisture conditions, and the competition between rainfall intensity and infiltration, may also contribute to this pattern."

We have specifically added the following content in the discussion:

"Additionally, the isotopic values of most groundwater in the gully areas are more depleted compared to those of rainfall and pond water, likely due to the recharge mechanisms and residence times of different groundwater types, and the inherent isotopic characteristics of their primary recharge sources (Ouali et al., 2024). **The depleted signatures in groundwater reflect preferential capture of isotopically light summer monsoon events, with effective percolation delayed to cooler seasons due to transient soil storage and minimized evaporation — consistent with observed water table rises predominantly from October to April.** Nevertheless, these values fall within the range of precipitation isotopic values, leaning towards the more negative end. This suggests two complementary mechanisms: (1) the thin unsaturated zone (<10 meters) provides preferential pathways for rapid infiltration of precipitation, minimizing evaporative fractionation, and (2) groundwater is likely recharged primarily by intense precipitation events (e.g., summer storms) with inherently more negative isotopic signatures.

Together, these processes explain the observed isotopic characteristics of groundwater."

7. The conceptual model (Fig. 10) emphasises the "restructuring" role of the gully system but does not discuss the potential risks of associated pollutant transport. Given that related issues are mentioned in the introduction, it is recommended to include a discussion on this aspect to present a more comprehensive perspective.

**Response:** Based on your comment, the potential risks of pollutant migration have been added to the discussion. It should be noted that, since this study does not involve the actual analysis of pollutant migration, the related content is discussed solely as background and future research directions. Therefore, the pollutant migration process is not explicitly represented in the conceptual model (Fig. 10) and is addressed only in the textual discussion. The specific content is as follows:

"However, with the reconstruction of gully systems and ecological restoration, attention must also be given to the potential risks of pollutant migration (Yu et al., 2020). The hydrological functions of gullies may enhance the movement of pollutants into groundwater, especially in areas with intensive human activities, where pollutants can enter gullies through surface runoff and subsequently infiltrate the groundwater system. During ecological restoration, excessive human intervention or soil improvement measures may lead to the accumulation and dispersion of pollutants, which may compromise groundwater security (Liu et al., 2017). Therefore, the protection and rational reconstruction of gully systems should not only focus on their hydrological functions but also consider potential environmental risks, particularly the pathways of pollutant migration."

**References**

Liu, Y.S., Chen, Z., Li, Y., Feng, W., Cao, Z. The planting technology and industrial development prospects of forage rape in the loess hilly area: A case study of newly-increased cultivated land through gully land consolidation in Yan'an, Shaanxi Province. Journal of Natural Resources. 32: 2065-2074, 2017.

Yu, Y.L., Jin, Z., Chu, G.C., Zhang, J., Wang, Y.Q., Zhao, Y.L. Effects of valley reshaping and damming on surface and groundwater nitrate on the Chinese Loess Plateau. Journal of Hydrology. 584: 124702, 2020.

8. The conclusion section (Section 7) provides a good summary of the study's core findings. However, some statements appear slightly absolute, such as the claim to be "the first to quantitatively identify the unique cascading recharge processes in a thin loess gully catchment" (Lines 781-782). While the research is innovative, caution is advised with phrases like "the first." It would be preferable to provide supporting literature references or adopt a more measured description.

**Response:** We have revised the relevant phrasing to address your concern. The specific revision is as follows:

"More importantly, these findings have direct implications for land management practices and ecological restoration strategies in similar arid regions worldwide. The study confirms that gullies serve as critical "recharge windows" for groundwater in arid areas. This underscores the importance of systematically identifying and conserving natural gully networks in watershed management, while avoiding excessive filling or hardening to preserve their hydrological functions. In ecological restoration projects, directing surface runoff toward gullies can efficiently convert limited precipitation into groundwater storage, thereby enhancing regional water retention capacity."

9. The manuscript is largely well-written, but some sections contain complex or awkward sentence structures that could be improved for readability. For instance, the introductory and results sections sometimes use dense scientific language, which might be simplified without losing technical precision. Additionally, the formatting of the references section could be revisited for consistency.

**Response:** Thank you for your positive assessment of the manuscript and for the constructive comments for improvement. We fully agree that enhancing clarity of expression and ensuring formatting consistency are essential for both readability and scientific rigor. In response to your comments, we have implemented the following comprehensive revisions:

First, we thoroughly reviewed the entire manuscript, with particular emphasis on the Introduction and Results sections, and systematically revised sentences with complex structures or awkward phrasing. While preserving scientific accuracy and completeness, we improved clarity

and fluency by breaking up long sentences, refining sentence structure, and optimizing the density of technical terminology.

Second, in accordance with the journal's guidelines, we carefully checked and standardized all in-text citations and the reference list to ensure full compliance. In addition, following your Specific Comment 10, we have incorporated the recommended key references into the manuscript.

We believe these targeted revisions have substantially improved the clarity, readability, and formatting consistency of the manuscript.

10. Some important references are missing from the introduction and discussion sections:

De Vries, J. J., & Simmers, I. (2002). Groundwater recharge: an overview of processes and challenges. Hydrogeology Journal, 10(1), 5-17.

Huang L.M., Shao M.A., Advances and perspectives on soil water research in China's Loess Plateau. Earth-Science Reviews, 2019: 102962.

Huang, L.M., Wang, Z.W., Pei, Y.W., Zhu, X.C., Jia, X.X., Shao, M.A., Adaptive water use strategies of artificially revegetated plants in a water-limited desert: A case study from the Mu Us Sandy Land. Journal of Hydrology, 2024, 644: 132103.

Xiang, W., Si, B. C., Biswas, A., & Li, Z. (2019). Quantifying dual recharge mechanisms in deep unsaturated zone of Chinese Loess Plateau using stable isotopes. Geoderma, 337, 773-781.

**Response:** We have carefully verified that the recommended references have been added or appropriately cited in the manuscript. We fully agree that including these important references significantly enhances the breadth and rigor of the study, and we have standardized the citation format in accordance with the journal's guidelines.

**References**

Huang L.M., Shao M.A. Advances and perspectives on soil water research in China's Loess Plateau. Earth-Science Reviews. 199(2): 102962, 2019. DOI: 10.1016/j.earscirev.2019.102962.

Huang, L.M., Wang, Z.W., Pei, Y.W., Zhu, X.C., Jia, X.X., Shao, M.A. Adaptive water use strategies of artificially revegetated plants in a water-limited desert: A case study from the Mu Us Sandy Land. Journal of Hydrology. 644: 132103, 2024. DOI: 10.1016/j.jhydrol.2024.132103.

Vries, J.J.D., Simmers, I. Groundwater recharge: an overview of processes and challenges. Hydrogeology Journal. 10(1): 5-17, 2002. DOI: 10.1007/s10040-001-0171-7.

Xiang, W., Si, B.C., Biswas, A., Li, Z. Quantifying dual recharge mechanisms in deep unsaturated zone of Chinese Loess Plateau using stable isotopes. Geoderma. 337: 773-781, 2019. DOI: 10.1016/j.geoderma.2018.10.006.

Xiang, W., Si, B.C., Biswas, A., Li, Z. Quantifying dual recharge mechanisms in deep unsaturated zone of Chinese Loess Plateau using stable isotopes. Geoderma. 337: 773-781, 2019. DOI: 10.1016/j.geoderma.2018.10.006.

---

## Author Comment (AC3)

**RC2**

We sincerely thank you for taking the time to review our manuscript and for providing valuable, professional, and rigorous feedback. Your constructive comments have been crucial in improving the quality of our manuscript. **Based on the comments from reviewers #RC1 and #CC1, we have addressed all your comments and, after careful reflection and multiple discussions, made comprehensive revisions, along with a detailed response to the revisions.** In response to your feedback, your comments are presented in red font, our responses in black, and the revisions to the manuscript in blue.

This manuscript addresses groundwater recharge processes in a gully system and aims to quantify recharge rates and pathways using hydrometric, isotopic, and geochemical approaches. While the topic is potentially interesting and relevant to HESS, the manuscript in its current form is poorly written, excessively long, and lacks a clear narrative structure. Moreover, the presentation of the results makes it difficult to assess whether the data adequately support the authors' conclusions. Interpretations are frequently motivated by background knowledge or earlier studies, yet the manuscript does not clearly distinguish between new insights derived from this work and those that primarily serve as contextual or corroborative information. This lack of separation between novelty and background substantially weakens the scientific message. I think substantial revision is required before the scientific contribution can be properly evaluated.

**Response:** We agree with your comments regarding the manuscript's structure and expression and have systematically revised it based on your general and specific comments to enhance its readability and academic rigor. The manuscript has been reorganized, redundant sections streamlined, and the logical flow between the Introduction, Methods, Results, and Discussion strengthened. The data analysis and interpretation in the Results section have been clarified to more effectively demonstrate the evidence supporting the conclusions. Additionally, the background information has been refined, with a clearer distinction made between existing research and new insights to highlight the uniqueness and contribution of our work. Below are the specific changes we have made, along with our point-by-point responses to your comments.

**General comments:**

1. A major issue is that Sections 1-3 contain extensive redundant descriptions, particularly regarding landscape characteristics, hill–gully contrasts, and background motivation. These sections mix site description, conceptual motivation, and literature background in a way that dilutes the main research questions and obscures the novelty of the study. As written, it is often unclear what information is background context, what is specific to the study site, and what directly supports the research objectives.

If I understand correctly, Sections 2-3 are primarily intended to function as a Materials / Study Site section, describing landscape structure and hydrological setting. However, the current version repeatedly interweaves general motivation (e.g., importance of gullies vs. hill) with sitespecific descriptions. This mixing weakens the paper's focus and makes the manuscript unnecessarily long. I would suggest the following structural changes:

• Condense Sections 1–3 substantially, removing repetitive explanations of hill vs. gully processes.

• Move most general motivation and background discussion to the Introduction.

• End the Introduction with a clear and concise paragraph that explicitly states why this study site was chosen, what relevant previous work has been conducted here, why this site is particularly suitable for addressing the stated research questions, and what the specific research questions or hypotheses are.

Response: Based on your valuable comments regarding the manuscript's structure, we have systematically revised and streamlined Sections 1 to 3 as follows:

➢ The original Section 2 has been removed, with its key points integrated into Sections 1 and 3.

➢ Redundant descriptions have been significantly condensed, especially in the comparison of hilly and gully processes.

➢ We have avoided the overlap between general research motivation and specific regional information, improving the coherence and clarity of the manuscript.

Additionally, we have explicitly added the scientific rationale for selecting the study area at the end of the Introduction, further clarifying the research questions and goals. These revisions aim to enhance the manuscript's logical focus and narrative clarity, while highlighting the novelty of the study. The specific revisions are as follows:

"**1. Introduction**

Groundwater recharge is a critical yet poorly understood component of hydrological cycles in dryland catchments (Li et al., 2024a). It is shaped by the precipitation regime, surface landcover heterogeneity, integrity of the subsurface regolith, characteristics of the underlying bedrock, and human interventions (Vries and Simmers, 2002; Owuor et al., 2016; Salek et al., 2018; Xu and Beekman, 2019; Zhang et al., 2020; Li et al, 2024b; Medici et al., 2024). While favorable subsurface flow pathways can locally enhance recharge, dryland regions are highly sensitive to even slight changes in precipitation, soil moisture, or runoff generation. This heightened sensitivity reflects their position along climatic ecotones and the influence of complex land–atmosphere–biosphere feedbacks (Kuang et al., 2019; Al-Oqaili et al., 2020; He et al., 2020; Jin et al., 2019; Jia et al., 2024). Small changes in these processes can cascade across catchments at various scales, amplifying existing vulnerabilities to ecological and social systems (Nicholson, 2011; Huang et al., 2017; Berg et al., 2016). In these fragile landscapes, understanding groundwater replenishment processes is crucial for sustaining ecosystems, securing water, and guiding restoration and management (Gleeson et al., 2016; Jasechko and Perrone, 2021; Scanlon et al., 2006).

Despite a growing body of research on groundwater recharge in (semi-) arid regions, significant knowledge gaps remain in landscapes with pronounced spatial heterogeneity, such as slopes, hilltops, and gully systems, where infiltration pathways and recharge processes can diverge sharply over short distances (Tooth, 2012; Manna et al., 2018; Letz et al., 2021). Gully systems, often seen as signs of land degradation, may beneficially act as recharge zones, capturing and infiltrating surface runoff during episodic rainfall (Tan et al., 2017; Li et al., 2024a; Xue et al., 2025). This same topographic focusing enables the rapid downslope transport of contaminants, including agricultural nutrients, sediments, and associated pollutants (Lian et al., 2025; Qu et al., 2025). However, the role of gullies in promoting vertical infiltration into groundwater is highly dependent on local subsurface connectivity and permeability conditions. Moreover, their broader hydrological functions remain poorly quantified, especially under the influence of widespread human interventions such as check dams and artificial ponds. While these structures are typically designed to arrest land surface degradation, they can substantially alter surface–subsurface connectivity and reshape recharge dynamics in uncertain ways (Lamontagne et al., 2021; Huang et al., 2019; Wang et al., 2023).

Worldwide, loess covers approximately 6% of the land surface area, forming discontinuous east–west belts in the mid-latitude forest-steppe, steppe, and desert-steppe zones of both hemispheres (Liu, 1985; Pécsi, 1990; Li et al., 2020). Among these, the Chinese Loess Plateau, the focus of our study accounts for approximately 7.4% of the global loess area (635,280 km²; Li et al.,

2020). It serves as a globally important natural laboratory for studying soil erosion and groundwater recharge processes, due to its exceptionally thick loess deposits (Li et al., 2021), highly erodible soils, intense summer rainstorms, and long history of agricultural activity, which collectively make it one of the most severely eroded regions worldwide (Shi and Shao, 2000; Fu et al., 2011). Its distinctive stratigraphic structure, characterized by thick, low-permeability loess layers, fundamentally governs groundwater behavior (Qiao et al., 2017). Meanwhile, extensive human interventions aimed at erosion control, including large-scale afforestation and gully engineering projects, have profoundly altered regional hydrological processes and the spatial redistribution of water (Wang et al., 2020; Zhao et al., 2024).

The setting for our investigation is a semi-arid landscape that has been shaped by severe soil erosion, extensively modified by engineered landforms; and it is now characterized by chronic water scarcity (Fu et al., 1999; Liu et al., 2017; Liu and Li, 2017; Li et al., 2021; Huang et al., 2024).

Water scarcity manifests as declining groundwater levels, reduced streamflow, dried-up wells and springs, and limited irrigation capacity (Yu et al., 2025). In such vulnerable environments, understanding the sources and sustainability of groundwater recharge is critical for long-term water resource management (Ajjur and Baalousha, 2021; Meles et al., 2024). Groundwater, for example, is a lifeline for rural communities in the hilly–gully region, yet scientific attention has largely bypassed the gullies themselves. Most previous studies have focused on recharge processes in tablelands and loess-covered hills, highlighting slow "piston flow" as the dominant mechanism (Huang et al., 2011, 2013; Li et al., 2017; Lu, 2020; Wang et al., 2024). However, the deep-profile recharge mechanisms observed in these areas may not apply to the gully-dominated landscapes of the Loess Plateau (Wang et al., 2024; Qiao et al., 2017; Zhu et al., 2018). Moreover, the hydrological functions of widely distributed gully systems, especially under the influence of engineering structures such as check dams, remain insufficiently quantified, and their underlying processes have long remained in the research shadow (Liu et al., 2011).

Therefore, this study selects the Nianzhuang Catchment, a typical gully area on the Loess

Plateau impacted by check dams, to establish a multi-method framework for assessing groundwater recharge by integrating stable isotope analysis ($\delta^2$H and $\delta^{18}$O), chloride concentrations, water table fluctuations, and hydro-statistical modeling. Specifically, our goals are to: (1) characterize the isotopic and hydrochemical signatures of precipitation, surface water (ponds), shallow pore water, and deeper fissure water; (2) identify and trace hydraulic connections and flow paths of different water bodies; and (3) quantitatively estimate pore-water recharge rates. This integrated approach aims to advance understanding of groundwater dynamics in complex dryland terrains, reframes engineered gully systems as critical recharge zones in engineered dryland landscapes, providing actionable insights for sustainable groundwater management and ecological restoration in the Loess

Plateau and similar semi-arid regions worldwide.

**133  2. Sampling site**

[revised manuscript text omitted]

Groundwater in the catchment can be broadly categorized into three types: pore water, spring water, and fissure water. Pore water is stored in permeable sandstone and conglomerate aquifers beneath loess and above mudstone or red clay. These aquifers are approximately 2–3 m thick, exhibit a sheet-like distribution, and have low water yield. Conceptually, "pore water" here refers to groundwater in a saturated aquifer, not to soil moisture. Fissure water occurs in fractured bedrock aquifers, which are spatially discontinuous due to irregular fracture development. The main water-bearing zones include cavities and jointed fissure networks, with an average aquifer thickness of about 6 m and moderate water yield. Hydraulic conductivity in these sandstone and conglomerate aquifers ranges from 0 to 0.47 m/d (Cai et al., 2019). Spring water emerges primarily at gully bases, especially in upper catchments, and originates from both pore and fissure sources, possibly supplemented by surface or pond water. Springs fed by pore water typically have low discharge rates (0–0.1 L/s) and low water yield, while those fed by fissure water exhibit moderate discharge rates (0.5–1.0 L/s) and moderate water yield.

Over recent decades, landscape rehabilitation through the Grain for Green Project and land reshaping under the Gully Land Consolidation Project have significantly altered the hydrological regime (Fu et al., 1999; Liu et al., 2017). Historically, surface runoff in the degraded catchment was flashy and episodic due to sparse vegetation. However, ecological restoration and small-scale engineering interventions, such as check dams, terraces, roads, and ponds, have moderated surface hydrology. Surface runoff, generated primarily during storm events, now contributes alongside delayed baseflow from groundwater recharge and interflow. The latter is often limited by the thick unsaturated zone in upland loess areas but may be enhanced in gully regions, where stratigraphy and land use favor infiltration (Wang et al., 2024; Gates et al., 2011). Gully areas also contain numerous check dams and ponds, with most water sourced from Hortonian overland flow of slope lands and direct rainfall. These small water bodies, often constructed for erosion control and water retention, influence local hydrological dynamics and may play a role in enhancing infiltration and recharge."

2. Interpretation and use of chloride concentrations. The role of chloride as supporting evidence for recharge pathways is repeatedly mentioned but remains vague and weakly justified. And I only found one figure in SI about chloride information, which is also not that informative as the author stated.

**Response:** To clarify, stable isotopes ($\delta^2H$ and $\delta^{18}O$) and chloride ions ($Cl^-$) are distinct tracers, each influenced by hydrological processes through different mechanisms. Stable isotopes are highly sensitive to evaporative fractionation, making them direct indicators for identifying water sources and evaporation history. In contrast, chloride ions generally exhibit conservative behavior during hydrological transport, with concentration changes primarily driven by physical mixing and evaporative concentration, without involvement in isotopic fractionation. This difference allows their combined use to provide more robust and comprehensive information for tracing water sources. In this study, chloride concentrations primarily support the isotope analysis, helping to validate water source mixing and groundwater recharge processes, and confirming that pore water is influenced by both precipitation and the mixing of precipitation with pond water.

Since chloride ions are not affected by fractionation during hydrological processes, their concentration changes are primarily driven by water source mixing and evaporation (water loss). Therefore, chloride plays a key role in resolving the "isotopic ambiguity" impacted by evaporation fractionation (open water). Our observational data show that the chloride concentration in pore water falls between that of low-concentration precipitation and high-concentration pond water. Additionally, the correlation between chloride concentration and $\delta^{18}O$ follows a conservative mixing model between precipitation and pond water. This evidence suggests that pore water chemistry changes are influenced by the mixing of chloride-rich pond water, reinforcing the mechanism of pore water recharge through the mixing of precipitation and surface water in the valley system.

Following your comments, we have moved the chloride concentration plot to the main text and added the correlation between chloride concentration and $\delta^{18}O$. This presents the key argument more clearly and comprehensively, thereby further enhancing the rigor and persuasiveness of our conclusions.

For example,

• Line 536: The statement that "multiple lines of observational evidence, including isotopic composition, chloride concentrations, and water age (ITTP)" support the identified pathways is too general. The manuscript does not clearly explain how chloride independently supports these conclusions.

**Response:** In the original manuscript, line 536 referred to observational evidence such as chloride concentrations and isotopic composition, aiming to provide multi-faceted support for the flow pathways identified by the SEM. To avoid presenting an oversimplified argument and to ensure that the discussion remains focused on the core results and interpretation of the SEM, we have removed this supplementary explanation in the revised manuscript, maintaining both coherence and academic rigor.

• Lines 547–552: The argument that similarities in chloride concentrations between pond water and pore water indicate mixed recharge is not logically developed. Chloride patterns alone do not necessarily imply source mixing without additional constraints (e.g., conservative behavior, spatial gradients, mass balance, or exclusion of evaporative concentration effects). The logic linking chloride distributions to the stated conclusions should be clarified and strengthened, or the claims should be toned down.

**Response:** Based on your comments, to strengthen the logical rigor of the conclusion regarding the similarity in chloride concentrations between pond water and pore water, we first confirmed the differences in chloride concentrations among various water sources. We then introduced the spatial relationship between $\delta^{18}O$ and chloride concentrations to further compare concentration variations across different water bodies at distinct locations. The results indicate a correlation between the distribution patterns of chloride concentrations and $\delta^{18}O$, providing additional support for the hypothesis of potential mixed recharge between pond water and pore water. The specific additions to the manuscript are detailed as follows:

"Complementing the isotope data, Cl⁻ levels in pore water consistently fall between those of precipitation and pond water across both seasons (Fig. 7a), and the correlation pattern between chloride concentration and $\delta^{18}O$ supports a mixed recharge origin for pore water (Fig. 7b). This trend aligns with the isotopic evidence from the rainy season and supports the interpretation that pond water contributes to pore water recharge via vertical percolation through the vadose zone, particularly during high-rainfall periods when infiltration capacity is exceeded.

[Figure]

Fig. 7. Chloride concentration of various water sources in the rainy and dry seasons (a), and the spatial relationship between chloride concentration and $\delta^{18}O$ for different water sources (b)."

• Line 856-858: The conclusion states "While isotopic evidence for recharge from pond water is obscured by evaporative fractionation, chloride concentrations provide a clear signal of subsurface connectivity." It is not supported by any direct or quantitative results presented in the manuscript. I

do not find clear evidence demonstrating such connectivity based on chloride data alone. Moreover, if chloride concentrations are intended to provide critical supporting information for the main conclusions, the relevant figure should be moved from the Supplementary Information to the main text, accompanied by a clearer and more rigorous explanation of how chloride constrains recharge pathways.

**Response:** Following your comment, we have provided further evidence of connectivity between pond water and pore water in the main text through both textual explanation and supplementary figures. Additionally, to ensure the conclusions are detailed and well-supported, we have revised the relevant section, with the specific revision as follows:

"Through integrated analysis of stable isotopes, chloride concentrations, water-table fluctuations, and inverse transit time proxies, this study provides multiple, convergent lines of evidence that engineered gully reaches on the Loess Plateau function as hydrologically significant recharge zones, rather than solely as products of accelerated erosion and degradation. Precipitation-driven runoff supports substantial recharge to shallow pore aquifers, with site-scale recharge magnitudes equivalent to approximately 43% of mean annual precipitation at the monitored gully reach.

Although evaporative fractionation limits the ability of stable isotopes alone to resolve direct recharge from ponded surface water, chloride concentrations provide independent evidence consistent with mixing between pond water and pore water, complementing the isotopic patterns.

Together, these indicators indicate likely hydraulic connectivity, while not constituting a mass- balanced quantification of recharge sources. Recharge within shallow gully-zone aquifers is spatially concentrated and temporally selective, governed by topographic convergence, loess stratigraphy, and ecological engineering structures, particularly check dams and ponds, which increase surface-water residence time and promote focused infiltration."

3. Role of surface water. The Discussion contains extensive statements regarding the large contribution of surface water to gully recharge. However, much of this discussion appears to rely on previous studies rather than direct analyses presented in this manuscript. The authors should clearly distinguish between conclusions derived from their own results, and contextual information drawn from earlier work.

**Response:** Thank you for this thoughtful comment regarding the role of surface water. One of the primary objectives of this study is to evaluate the contribution of surface water—represented mainly by pond water—to groundwater recharge in gully systems. Using stable isotope data ($\delta^2$H and $\delta^{18}$O)

together with chloride concentrations, we provide direct evidence in the Results section for hydraulic linkage between pond water and pore water. This linkage is further quantified using the structural equation model (SEM), which explicitly evaluates recharge pathways and their relative strengths. The SEM results indicate that the direct effect of pond water on pore water is significantly stronger than that of precipitation, suggesting that pond water acts as an important intermediary in the recharge process within the study catchment.

At the same time, we recognize that parts of the Discussion refer to broader hydrological processes that have been documented in previous studies. Following your suggestion, we have carefully revised the Results and Discussion sections to clearly distinguish between conclusions that are directly supported by our data and analyses, and contextual interpretations that are informed by earlier work. Conclusions derived from this study are now explicitly attributed to our observations and modeling results, whereas references to more general gully hydrological functions or the impacts of engineering measures are clearly framed as supporting background. These revisions help clarify the evidentiary basis of our conclusions and strengthen the overall rigor of the manuscript. The main revisions are as follows:

"In recent years, discussions of groundwater recharge sources on the Loess Plateau have largely focused on tableland and hilly areas characterized by thick loess deposits, whereas gully regions have received comparatively limited attention (Li et al., 2017; Xiang, 2020; Lu, 2020). For instance, Liu et al. (2011) demonstrated that groundwater near valley bottoms in hilly loess areas can be replenished by a combination of precipitation, runoff, and surface water. Our results are broadly consistent with these earlier findings, but extend them by providing multiple lines of site-specific evidence. Based on stable isotope signatures and chloride concentrations, we independently identify precipitation and surface water as the primary sources of groundwater recharge in gully systems. Furthermore, by applying a structural equation model (SEM), we quantitatively evaluate the relative importance of different recharge pathways, demonstrating that surface water (particularly pond water) plays a key mediating role in transferring precipitation inputs to subsurface pore water. Building on these results, we classify groundwater in the study area into three functional types, spring water, pore water, and fissure water, and propose a progressive, multi-stage recharge framework: (1) direct recharge of pond water by precipitation and indirect recharge of pore water by precipitation; (2) focused recharge from pond water to pore water; and (3) downward percolation from pore water to fissure water. This framework highlights the complexity of groundwater flow and recharge processes in gully-dominated landscapes and underscores the significant influence of human interventions, such as ponds and check dams, on modifying hydrological connectivity and recharge dynamics."

"This conceptual reframing is grounded in the stark hydrological contrasts between hilly uplands and gully systems and directly addresses a critical knowledge gap in understanding the hydrological functioning of managed gully environments. In the hilly uplands, previous studies have shown that thick loess deposits, often exceeding 90 m (including low-permeability aquifers), combined with steep slopes (>15°) severely restrict vertical infiltration (Zhu et al., 2018; Huang et al., 2019; Huang et al., 2024). Compounded by short-duration, high-intensity rainfall events that provide insufficient moisture for deep profile wetting, this results in the rapid conversion of rainfall into surface runoff (Li et al., 2021). This study further clarifies that the runoff is systematically funneled downslope into gully systems as a consequence of ecological engineering interventions, such as check dams and retention ponds that intercept and concentrate overland flow. Most infiltration occurs after surface water accumulates in engineered gullies, particularly within perched water bodies like ponds, which subsequently serve as localized recharge foci, a conclusion supported by the isotopic and hydrochemical evidence presented in this study."

4. Hill versus gully. The results presented in this study are derived exclusively from the gully system, and the manuscript does not include a direct comparison of recharge behavior between hill and gully settings at the same site and during the same period. As such, the authors should be very cautious in how they frame both the Introduction and the Conclusions, particularly where broader contrasts between hill and gully recharge processes are implied. Given the absence of contemporaneous hillslope observations, statements suggesting relative differences in recharge magnitude or pathways between hill and gullies should be clearly identified as inferences based on previous studies, rather than findings derived from the present work. This distinction is especially important in the conceptual framework and schematic figures, where hill processes appear alongside gully processes without sufficiently clear attribution. One example is the conceptual figure (Fig. 10). I recommend that the authors:

• Explicitly state which components or pathways are supported by results from this study and which are drawn from previous literature;

• Redraw the figure to include quantitative or semi-quantitative information (e.g., relative magnitudes, ranges, or percentages of pathways) where supported by data.

In its current form, the conceptual figure does not clearly highlight new insights generated by this study, and instead risks reinforcing a narrative largely based on prior work.

**Response:** Thank you very much for your constructive comment. In response to your General comment #1, we have thoroughly revised the Introduction section, emphasizing the novelty and scientific significance of groundwater recharge processes in gully areas under engineering interventions. This study specifically focuses on hydrological processes within the valley zone and does not directly address hillslope hydrology. When referring to the hilly area, we have positioned the hillslope solely as a contributing source of runoff into the valley, drawing on previous study findings and our own field observations. The core framework of this study can be summarized as follows: surface runoff from the upland hillslope converges into the gully, where it is intercepted by check dams, forming pond storage that subsequently recharges groundwater.

Following your comment, we have redrawn Fig. 10 (in the original manuscript, and now Fig. 11 in the revised manuscript) to clearly define the spatial scope of this study as the gully area, with specific annotations for clarity. Accordingly, we have systematically reviewed and revised the Discussion section to ensure that all analyses, inferences, and conclusions are tightly focused on the hydrological processes within the gully area. The revisions are as follows:

"**5.4. Revised conceptual model**

To convey our evolving understanding of the spatial structure and dynamics in the Gully Region, we developed a conceptual model that reframes engineered gully systems not simply as erosion features but as hydrologically active conduits for groundwater recharge (Fig. 11). This framework traces precipitation's transformation into subsurface water, from runoff capture and surface ponding in dammed gully reaches, through infiltration in the unsaturated zone, to recharge in both shallow porous aquifer and deeper bedrock fissure systems.

This conceptual reframing is grounded in the stark hydrological contrasts between hilly uplands and gully systems and directly addresses a critical knowledge gap in understanding the hydrological functioning of managed gully environments. In the hilly uplands, previous studies have shown that thick loess deposits, often exceeding 90 m (including low-permeability aquifers), combined with steep slopes (>15°) severely restrict vertical infiltration (Zhu et al., 2018; Huang et al., 2019; Huang et al., 2024). Compounded by short-duration, high-intensity rainfall events that provide insufficient moisture for deep profile wetting, this results in the rapid conversion of rainfall into surface runoff (Li et al., 2021). This study further clarifies that the runoff is systematically funneled downslope into gully systems as a consequence of ecological engineering interventions, such as check dams and retention ponds that intercept and concentrate overland flow. Most infiltration occurs after surface water accumulates in engineered gullies, particularly within perched water bodies like ponds, which subsequently serve as localized recharge foci, a conclusion supported by the isotopic and hydrochemical evidence presented in this study.

Crucially, gully systems possess distinct hydrogeological characteristics: the loess mantle is much thinner (typically < 25 m), and the soils are dominated by silt loam textures with moderate specific yield (0.02–0.05) and high field capacity (21–28%). These properties promote transient water storage and enable temporally delayed and depth-partitioned infiltration. Based on our integrated analyses of stable isotopes, chloride concentrations, and inverse transit time proxies, we find that engineered gullies function not as passive erosional features but as active, managed recharge conduits. This conceptualization captures a critical spatial transition, from runoff generation in the hilly uplands to focused recharge in gully zones, emphasizing the pivotal role of gully systems in regulating groundwater recharge across the Loess Plateau landscape.

Combined hydrological monitoring and multi-indicator analysis further reveal that following the rainy season, infiltration depths on hilly slopes are typically shallow (less than 1 m), while groundwater levels in gully areas exhibit pronounced rises exceeding 2 m (Fig. 11). Recharge estimates based on the water table fluctuations reach up to approximately 240 mm at the monitored gully reach, far surpassing values observed in deep unsaturated zones of tablelands and hills (Huang et al., 2011; Li et al., 2017; Lu, 2020; Wang et al., 2024). The results of this study reinforce the role of engineered gully reaches as focal points for groundwater recharge and further quantify site-scale pore-water recharge equivalent to ~43% of mean annual precipitation, a finding that highlights the efficiency of focused infiltration under managed conditions.

Liu et al. (2011) found that groundwater near valleys in the hilly loess area is replenished by precipitation, runoff, and surface water. Moreover, fissure water exhibits more depleted isotopic signatures and higher chloride concentrations, indicating deeper percolation of pore water or mixing with older recharge sources (Fig. 11). These patterns, supported by ITTPs and statistical (SEM-based) connectivity indicators, reveal a hierarchical recharge sequence: event-driven infiltration enters a porous shallow aquifer, some of which slowly percolates into deeper fissure zones. This hierarchical mechanism is facilitated by the combination of thin loess mantles, engineered interventions (e.g., check dams and ponds), and delayed hydrological responses.

By integrating multiple lines of evidence, this conceptual model redefines engineered gullies as selective recharge corridors whose hydrological function emerges from the interaction between geomorphic structure and human intervention. It challenges the traditional view of gullies as purely erosional landforms and emphasizes their dual hydrological function: acting both as runoff conveyance channels and as transient reservoirs that store and redistribute water across space and time. This recharge capacity is jointly governed by topographic convergence, reduced loess thickness, and the presence of engineered structures such as check dams and retention ponds that increase residence time.

Crucially, the model offers insight into the multifunctionality of ecological engineering, particularly check dams and ponds, in enhancing groundwater recharge, and supporting ecosystem restoration across the Loess Plateau. This study proposes a cascade-type recharge framework for engineered gully systems, highlighting the role of engineered gullies as convergence pathways that locally focus infiltration and groundwater recharge. Rather than invoking preferential flow within the soil matrix, this framework emphasizes topographic convergence, stratigraphic thinning, and engineered ponding as the dominant mechanisms that promote spatially concentrated recharge within gully zones. While this process is demonstrated using site-specific tracer and water-table observations, its broader relevance at the catchment scale remains conceptual and warrants further investigation. Furthermore, water movement within the silted loess layer of the gully system remains dominated by a piston flow pattern (Yu et al., 2025). By identifying the pivotal role of gully systems in stormwater detention, delayed infiltration, and depth-partitioned recharge, this study establishes a mechanistically grounded conceptual basis improving water resource allocation, infrastructure planning, and groundwater sustainability in arid and semi-arid regions.

However, with the reconstruction of gully systems and ecological restoration, attention must also be given to the potential risks of pollutant migration (Yu et al., 2020). The hydrological functions of gullies may enhance the movement of pollutants into groundwater, especially in areas with intensive human activities, where pollutants can enter engineered gullies through surface runoff and subsequently infiltrate the groundwater system. During ecological restoration, excessive human intervention or soil improvement measures may lead to the accumulation and dispersion of pollutants, which may compromise groundwater security (Liu et al., 2017). Therefore, the protection and rational reconstruction of gully systems should not only focus on their hydrological functions but also consider potential environmental risks, particularly the pathways of pollutant migration.

These findings therefore underscore the need to evaluate gully-based restoration strategies within an integrated water-quality and groundwater-protection framework.

The study confirms that hydrologically arrested gully systems can function as critical "recharge windows" for groundwater in arid areas. This underscores the importance of strategically identifying and managing gully networks in watershed management, while avoiding excessive filling or hardening to preserve their hydrological functions. In ecological restoration projects, directing surface runoff toward engineered gullies under controlled conditions can efficiently convert limited precipitation into groundwater storage, thereby enhancing regional water retention capacity. Beyond advancing theoretical understanding of regional hydrological processes, this conceptual model provides a process-based foundation for developing spatially targeted models of groundwater recharge in managed dryland landscapes.

[Figure]

Fig. 11. Hydraulic connections between different water bodies in the hilly-gully region of the Loess

Plateau. The study area consists of hilly and gully regions. In the hilly area, the stratigraphic sequence from top to bottom is Malan loess, Lishi loess, red clay, sandstone, and mudstone. Rainfall infiltration within the Malan loess is less than 1 m, and the area is mainly covered by vegetation. In the gully area, the stratigraphy from top to bottom includes loess (silt), sandstone and conglomerate, and mudstone. Pore water is found within the sandstone and conglomerate, while fissure water occurs in bedrock fractures (mudstone). Numerous check dams or ponds are distributed throughout the gully area. The vertical separation between the pore water and pond water ranges from 3 to 5 m. Corn is the main crop cultivated in this region. Most springs in the study area are located at the junction of the hilly and gully regions and are discharged from pore water."

**Specific comments:**

1. Fig. 1: Please label the horizontal and vertical scale of the hillslope profile. Without scale information, the geomorphic interpretation is unclear. And consider to switch the order of Fig. 1 and 2.

**Response:** Following your comment, we have added clear scale information to the hillslope profile. Specifically, both horizontal and vertical scale bars have been included to ensure a clear and accurate interpretation of the geomorphological features. The revised Fig. 2 (in the revised manuscript) is as follows:

[Figure]

Fig. 2. The topographic profile of the Nianzhuang Catchment in the hilly region of the Loess Plateau. Full profile from the top to mid-slope (a); two repeated mid-slope profiles (b, c). The photo was taken after a 41 mm rainfall event over four days. Subsequent measurements showed that infiltration depths reached only 20–30 cm at the top of the slope, compared to approximately 80 cm at the mid-slope positions.

In response to General comment #1, we have relocated Fig. 1 to the "2. Sampling Sites" section and swapped the order of Fig. 1 and 2. This adjustment ensures that the figures are arranged logically to align with the structure of the section content.

2. Lines 272–273: The relationship between groundwater level and water pressure is introduced without sufficient justification. Why were these parameters selected over others? Please clarify the physical reasoning.

**Response:** According to the principles of hydrostatics, the hydrostatic pressure $P$ at the sensor is related to the height $h$ of the overlying water column by $P=\rho gh$, where $\rho$ is the water density and $g$ is the gravitational acceleration. In an unconfined aquifer, the pressure measured by the sensor corresponds to the hydrostatic pressure exerted by the overlying water column. This allows for the calculation of the water column height $h$, and, combined with the sensor's elevation, the depth to the groundwater table can be determined. This method, based on the classical hydrostatic equilibrium principle, is a standard hydrological monitoring technique with a solid physical foundation and reliable measurement accuracy. Relevant content has been added to the manuscript, as detailed below:

"Precipitation was collected from October 24, 2023, to October 24, 2024, using a weather station situated in an open field within the catchment. Continuous groundwater level data were recorded from September 24, 2023, to December 20, 2024. **Groundwater pressure and temperature were monitored using Onset HOBO U20-001-03 sensors (20 m range), with a pressure accuracy of ±0.3% FS (±2.55 kPa) and a resolution of <0.085 kPa, and a temperature accuracy of ±0.44 °C with a resolution of 0.1 °C. The sensor was calibrated to atmospheric pressure before installation to ensure accurate measurement of absolute static water pressure, and water table levels were calculated based on the measured pressure data.** The conversion relationship between water pressure and groundwater level is given by $Y = 0.86 \times X - 22.1$ where $Y$ represents the groundwater level and $X$ represents the water pressure. **The conversion between water pressure and groundwater level is based on the principle of hydrostatics. The hydrostatic pressure $P$ at the sensor is related to the height of the overlying water column $h$ by $P=\rho gh$, where $\rho$ is the water density and $g$ is the gravitational acceleration. In unconfined aquifer, the pressure measured by the sensor corresponds directly to the static pressure exerted by the overlying water column. From this, the water column height $h$ can be calculated, and combined with the sensor's installation elevation, the depth to the groundwater table can be determined.** Notably, the monitoring well is located in the pore water layer of the gully region. The well is hand-dug (1.1 m wide, 10 m deep) and is unaffected by human activities."

3. Lines 428–430 / Fig. 4c: Fig. 4c does not show a consistently decreasing trend of specific yield with depth. The statement that "Specific yield (Sy) peaks at −20 cm (4.5%) but decreases with depth" is not convincingly supported by the figure. The interpretation that deeper layers "store water with minimal drainage" therefore appears overstated and should be revised or better supported.

**Response:** Following your comment, we have revised the figure captions. The specific revisions are as follows:

[Figure]

Clay (%)    Silt (%)    Sand (%) —⊕— FWC (%) —⊕— TP (%) —⊕— Sy (%)

Fig. 4. Vertical variation in soil texture and water retention characteristics in the gully region of the

Loess Plateau. (a) Soil particle size distribution by depth, showing relatively uniform composition across layers (10–50 cm), dominated by silt (64–65%), with moderate clay (16–20%) and low sand (16–20%) content. This fine-textured profile supports high moisture retention and slows infiltration, promoting delayed recharge. (b) Depth profiles of total porosity (TP) and field water capacity (FWC)

reveal decreases with depth to 40 cm, with FWC reaching ~27%, suggesting greater water-holding capacity in subsoil layers and enhanced buffering of infiltrated water. **(c) Vertical variations in the**

**Specific Yield (Sy) across different soil layers.** Collectively, these physical properties reflect a vertically stratified soil system where near-surface layers regulate infiltration pulses, and deeper layers act as long-term storage, shaping the timing and magnitude of subsurface recharge.

4. Fig. 5: The current representation of rainy versus dry seasons is unclear. The figure does not effectively illustrate isotopic differences between seasons, making the associated text difficult to support. Presenting seasonal mean values (or distributions) for each water type would likely convey the message more clearly.

**Response:** We agree with your comment that Fig. 5 provides relatively limited information on isotopic data for the wet and dry seasons. To more systematically and comprehensively present the seasonal characteristics of isotopic values across various water bodies, we have supplemented the data in Fig. 6 and Table A2 in the original manuscript. Specifically, the box plots in Fig. 6 visually display the distribution range, median, and variability of $\delta^{18}O$ and $\delta^2H$ for each water source during both wet and dry seasons, facilitating comparison of overall seasonal differences and variation patterns. Table A2 provides statistical metrics, such as mean values and standard deviations, for each water type's isotopes during both seasons, enabling a quantitative comparison. The specific details are as follows:

[Figure]

Fig. 6. Dual stable isotopic compositions of rainfall, pond water, spring water, pore water, and fissure water during the rainy season and dry season in the gully region of the Loess Plateau. The black line represents the global meteoric water line (GMWL, $\delta^2H=10+8\delta^{18}O$). GMWL is the global meteoric water line of Craig, LMWL is the local meteoric water line, SWL is the spring water line,

POWL is the pond water line, FWL is the fissure water line, and PWL is the pore water line. Panels (b) and (d) are magnified views of (a) and (c), respectively, highlighting the isotopic compositions of pore water, fissure water, and spring water (x-axis: −12 to −6‰; y-axis: −80 to −50‰).

Table A2. Isotopic composition ($\delta^2$H and $\delta^{18}$O) of various water sources in the rainy and dry seasons

| | Rainy season | | Dry season | |
|---|---|---|---|---|
| | $\delta^2$H | $\delta^{18}$O | $\delta^2$H | $\delta^{18}$O |
| Rainfall | −36.6±20.4‰ | −5.6±2.3‰ | −31.0±23.2‰ | −4.9±3.0‰ |
| Pond water | −40.5±13.1‰ | −4.1±2.3‰ | −24.5±6.9‰ | −0.8±1.3‰ |
| Spring water | −67.3±2.6‰ | −9.0±0.4‰ | −68.4±2.2‰ | −9.0±0.4‰ |
| Pore water | −66.3±3.1‰ | −9.0±0.6‰ | −65.4±3.8‰ | −8.5±0.6‰ |
| Fissure water | −65.0±3.8‰ | −8.8±0.9‰ | −64.5±5.5‰ | −8.5±0.9‰ |

5. Fig. 8: The meaning of "direct effects" and "total effects" is not clearly explained. Please clarify these terms explicitly in the caption and main text.

**Response:** Following your comment, we have added explanations of "direct effects" and "total effects" in the figure caption and methods section of the manuscript.

"Structural Equation Modeling (SEM) has been widely applied in water science to evaluate complex relationships among hydrological, geological, and anthropogenic variables, particularly in studies of groundwater contamination and water quality degradation (Wu, 2010; Lupi et al., 2019; Xie et al., 2025). In this study, SEM is used explicitly as an exploratory, hypothesis-generating tool to assess potential hydrological connectivity among water sources based on dual-isotope ($\delta^2$H–$\delta^{18}$O)

data from rainfall, pond water, spring water, pore water, and fissure water. SEM is not a mass- conserving or process-based flow model, nor is it used here to infer volumetric fluxes, recharge rates, or source apportionment. Instead, it serves as a statistical consistency check on hypothesized connectivity, identifying direct and indirect associations among water bodies that are evaluated in conjunction with tracer evidence and hydrometric observations.

Within the SEM framework, path relationships are primarily explained through two types of effects: The direct effect refers to the immediate impact of one variable on another through a single path, typically quantified as a standardized regression coefficient. Total effect represents the overall

impact of one variable on another through all possible paths (including both direct and indirect),

calculated as the sum of the direct effect and all indirect effects. Comparing direct and total effects

allows identification of intermediary linkages and dominant association structures within the

"Fig. 9. Structural equation modeling (SEM) and variance partitioning results illustrating hydraulic connectivity among water sources in the gully region of the Loess Plateau. Panels (a) and (b) show the standardized direct (a) and total effects (b) among rainfall, pond water, pore water, spring water, and fissure water, based on $\delta^{18}O$ and $\delta^2H$ data. **In SEM, the total effect includes both direct**

**pathways (a; e.g., rainfall → pore water) and indirect pathways mediated by other variables**

**(b; e.g., rainfall → pond water → pore water).** Arrows indicate hypothesized water flow pathways, with line thickness proportional to effect size. Asterisks denote statistical significance (*$P < 0.05$,

**$P < 0.01$, ***$P < 0.001$). The model fit is excellent ($\chi^2 = 0.3$, df = 2, RMSEA = 0.009, CFI = 1.0,

NFI = 0.994), supporting the robustness of these inferred connections. Panels (c) and (d) present variance partitioning results showing the relative contributions of source waters to pore water and fissure water during the rainy and dry seasons, respectively. In panel (c), rainfall (red) and pond water (pink) explain a large portion of pore water variability, with some shared explanatory power and modest residuals. In panel (d), fissure water reflects a more complex origin, with contributions from rainfall (red), pond water (pink), and pore water (blue), and greater overlap and residuals, especially during the dry season."

6. Fig. 9: The lines representing the "RISE" and "MRC" methods are not clearly distinguishable in the figure.

**Response:** In the original manuscript, the "RISE" and "MRC" curves were plotted on the same axis to facilitate a direct comparison of their results. As you rightly observed, the close similarity between the two methods made the lines difficult to distinguish, which compromised the clarity and effectiveness of the information presented.

Based on your comment, we have redrawn and optimized Fig. 9 (in the original manuscript, and now Fig. 10 in the revised manuscript) and revised its caption accordingly. The specific revisions are as follows:

[Figure]

Fig. 10. Temporal dynamics of pore water table depth, temperature, precipitation, and recharge in the gully region of the Loess Plateau. (a) Daily time series of pore water table depth (blue line) and surface temperature (red line) from September 2023 to November 2024. The water table fluctuates seasonally, rising from ~−8.1 m in late summer to a maximum of ~−5.0 m in early spring (March

2024), indicating delayed infiltration and cool-season recharge. **(b) Daily precipitation (blue bars)**

**and modeled pore water recharge estimates using the MRC methods. (c) Daily precipitation**

**(blue bars) and modeled pore water recharge estimates using the RISE methods.** Most recharge events occur from October to April, even when rainfall is not especially high, while warm-season precipitation contributes little to recharge, likely due to increased evaporative losses and shallow soil retention. Together, these patterns suggest strong seasonal control on recharge processes, with effective infiltration primarily occurring during cooler, low-evaporation periods.

---

## Author Comment (AC4)

**RC1**

We are grateful for your thoughtful and constructive comments, which have provided invaluable guidance in strengthening our work. **In response to your feedback and the comments from reviewers #RC2 and #CC1, we have thoroughly revised the manuscript, and this is our second reply to your valuable comments.** Your comments are presented in red font, our responses in black, and the revisions to the manuscript in blue.

This multidisciplinary study on the Loess Plateau centers on surface–groundwater interactions and fits well within the scope of the Journal-HESS. Based on extensive field observations, the manuscript investigates groundwater recharge processes within gully systems on the Loess Plateau, aiming to reframe gullies as hydrologically active recharge zones rather than merely erosional landforms. The study uses an integrated, multi-method approach—including stable isotopes, chloride concentrations, water table fluctuation (WTF) analysis, and structural equation modeling (SEM)—to examine the linkages among precipitation, surface water, and different groundwater bodies. The authors have invested substantial effort in data collection, fieldwork, and laboratory analyses. Given the increasing importance of groundwater sustainability in arid regions, the study carries clear novelty and relevance, and makes several notable contributions: (1) Reframing the hydrological role of gullies in the loess hilly region (core innovation); (2) Identifying the key mechanisms and process chain of groundwater recharge within gully systems; (3) Demonstrating the significant enhancement of groundwater recharge by engineering interventions (check dams and ponds). Overall, the manuscript is of good quality but still lacks certain details. The following specific comments may help strengthen the paper. I recommend publication after moderate revision.

**Response:** Thank you for taking the time to review our manuscript and for providing valuable and constructive comments. Your feedback has greatly helped us improve the manuscript. We fully agree with your comments and have made substantial revisions to enhance its readability and academic rigor. Below are the specific changes we have made, along with our point-by-point responses to your comments.

1. Line 85-90: Clearly state the research goals to fully encompass the study content. It is recommended to include a goal specifically addressing the analysis of isotopic characteristics, which will ensure alignment with your methodology and results.

**Response:** We fully accept your comment and have revised the research goals to include a specific focus on isotopic characteristics, ensuring alignment with the methodology and results. The specific revision is as follows:

"Therefore, this study selects the Nianzhuang Catchment, a typical gully area on the Loess Plateau impacted by check dams, to establish a multi-method framework for assessing groundwater recharge by integrating stable isotope analysis ($\delta^2H$ and $\delta^{18}O$), chloride concentrations, water table fluctuations, and hydro-statistical modeling. Specifically, our goals are to: (1) characterize the isotopic and hydrochemical signatures of precipitation, surface water (ponds), shallow pore water, and deeper fissure water; (2) identify and trace hydraulic connections and flow paths of different water bodies; and (3) quantitatively estimate pore-water recharge rates. This integrated approach aims to advance understanding of groundwater dynamics in complex dryland terrains, reframes engineered gully systems as critical recharge zones in engineered dryland landscapes, providing actionable insights for sustainable groundwater management and ecological restoration in the Loess Plateau and similar semi-arid regions worldwide."

2. Line 85-90: Rearrange the research goals to align with the structure of the results section, as the order of goals 1 and 2 appears to be reversed. The goals should follow the sequence in which the results are presented.

**Response:** We agree with your comment. The specific revision is as follows:

"(1) characterize the isotopic and hydrochemical signatures of precipitation, surface water (ponds), shallow pore water, and deeper fissure water; (2) identify and trace hydraulic connections and flow paths of different water bodies; and (3) quantitatively estimate pore-water recharge rates."

3. Line 180: How many wells in this catchment were monitored? Please show their positions in Figure 2.

**Response:** The monitoring network in this study includes 35 discrete sampling wells: 9 for pore water, and 26 for fissure water, aimed at characterizing the spatial variability of groundwater hydrochemistry and isotopic signatures. All sampling locations are clearly marked in Fig. 2 (in the original manuscript, and now Fig. 1 in the revised manuscript) of the original manuscript.

Additionally, one continuous pore water table monitoring well is installed in the middle reaches of the catchment to quantify groundwater table fluctuations and estimate recharge rates. In response to your comment, we have updated Fig. 2 (in the original manuscript, and now Fig. 1 in the revised manuscript) to include the location of this monitoring well. The specific revisions are detailed below.

[Figure]

Fig. 1. The geographical location and sampling sites for rainfall, pond water, pore water, spring water, and fissure water in the Nianzhuang catchment. The Nianzhuang catchment is located in the hilly and gully region of the central Loess Plateau, with elevations ranging from 896 to 1269 m. The average depth of pore water wells is $8.0 \pm 1.5$ m (range: 4–10 m), while that of fissure water wells is $57.6 \pm 29.2$ m (range: 25–170 m). These sampling sites represent locations where both rainy and dry season samples were collected, and are all situated within the gully areas of the catchment.

4. It is recommended to unify the units in Line 235 ('m/day') and Line 214 ('m/d') for consistency.

In addition, Line 213-214, the permeability of Neogene coarse sandstone and conglomerate here couldn't possibly be this (7.5–36.19 m/d) high. I suspect the authors might have made a mistake with the units. Please double-check.

**Response:** We fully agree with your comment and have made the necessary revisions. In the original manuscript, the permeability unit at Line 214 was listed as 'Lu' but was incorrectly noted as 'm/d'

(7.5–36.19 Lu ≈ 0.07–0.31 m/d). In this revision, all permeability units have been uniformly converted to the standard unit 'm/d' based on the conversion relationship and applied consistently throughout the manuscript. The specific revisions are detailed below.

"The significant reduction in loess thickness, combined with the relatively high permeability of

Neogene coarse sandstone and conglomerate (0.07–0.31 m/d), creates favorable conditions for infiltration and focused recharge."

5. Line 494: The numbers in the global meteoric water line equation need to be superscripted.

**Response:** We have made the necessary revisions as per you commented, and the numbers in the global meteoric water line equation have now been superscripted. Thank you for pointing this out.

6. Line 614-629: When explaining the phenomenon that 'the isotopic values of most groundwater in the gully areas are more depleted compared to those of rainfall and pond water', ensure the logical connection between 'the thin unsaturated zone' and 'direct recharge from intense rainfall events' is fully articulated, and consider including a discussion on the 'seasonal precipitation isotope effect.

**Response:** We fully agree with your comment and have made the necessary revisions to clarify the logical connection between "the thin unsaturated zone" and "direct recharge from intense rainfall events". Additionally, we have included a discussion on the "seasonal precipitation isotope effect"

to further enhance the explanation. The specific revisions are as follows:

"Additionally, the isotopic values of most groundwater in the gully areas are more depleted compared to those of rainfall and pond water, likely due to the recharge mechanisms and residence times of different groundwater types, and the inherent isotopic characteristics of their primary recharge sources (Ouali et al., 2024). The depleted signatures in groundwater reflect preferential capture of isotopically light summer monsoon events, with effective percolation delayed to cooler seasons due to transient soil storage and minimized evaporation, consistent with observed water table rises predominantly from October to April. Nevertheless, these values fall within the range of precipitation isotopic values, leaning towards the more negative end. This suggests two complementary mechanisms: (1) the thin unsaturated zone (<10 meters) provides preferential pathways for rapid infiltration of precipitation, minimizing evaporative fractionation, and (2)

groundwater is likely recharged primarily by intense precipitation events (e.g., summer storms) with inherently more negative isotopic signatures (Liu et al., 2024). Together, these processes explain the observed isotopic characteristics of groundwater."

**Reference**

Liu, Y.Z. Source analysis of precipitation chemical components on the Loess Plateau based on hydrogen and oxygen stable isotopes[D]. Northwest A&F University, 2024.

DOI:10.27409/d.cnki.gxbnu.2024.001528.

7. Line 657-666: When describing the differences between previous studies and this research, it is essential to explicitly highlight the fundamental distinctions in 'spatial scale' and 'hydrological units'

to more precisely define the original contribution of your work.

**Response:** We fully agree with this insightful suggestion and have revised the manuscript to explicitly highlight the fundamental distinctions in spatial scale and hydrological units between previous studies and our work. The specific revisions are as follows:

"In summary, while hillslope-scale studies describe a "dispersed recharge" mode, where precipitation percolates slowly through thick unsaturated zones, this study identifies a "concentrated recharge" mode in engineered gullies, driven by runoff convergence and regulated by check dams via ponding. These fundamentally distinct modes, differing in hydrological processes, spatial scales, and recharge efficiencies, collectively enhance the understanding of groundwater recharge mechanisms on the Loess Plateau."

8. Line 807-815: When addressing the limitations of isotopes and structural equation modeling, the advantages of the Water Table Fluctuation (WTF) method should be articulated more precisely.

Emphasize that these methods are 'complementary' rather than 'contradictory', thus presenting a more balanced argument.

**Response:** We fully agree with this constructive suggestion. To present a more balanced and precise argument, we have revised the relevant section to explicitly present the multi-method approach as complementary, rather than contradictory. The specific revisions are as follows:

"Without explicit mass-balance constraints, structural equation modeling may not independently or quantitatively represent actual groundwater flow processes. In contrast, the water-table fluctuation method, which directly measures changes in groundwater levels, provides a more empirically grounded estimate of total recharge. Each approach nevertheless offers distinct strengths: water- table fluctuations resolve the timing and magnitude of recharge, whereas isotopic, hydrochemical, and modeling analyses yield critical insights into recharge sources and flow pathways. By leveraging the complementarity and mutual corroboration of these methods, our study robustly demonstrates the pivotal role of gully areas in groundwater recharge."

**Figures and tables**

1. Fig. 3a and 3c lack units on the x-axis. Additionally, the directions of profiles Line1 and Line2

should be clearly indicated in Fig. 3b.

**Response:** We appreciate your comment and have made the necessary revisions. The units on the x-axis have been added to both Fig. 3a and 3c. Additionally, the directions of the profiles Line1 and

Line2 have been clearly indicated in Fig. 3b. The specific additions are as follows:

[Figure]

① Loess (silt) ② Malan loess ③ Lishi loess ④ Red clay ⑤ Mudstone ▮ Pond water
⑥ Coal seam ⑦ Sandstone, conglomerate ⑧ Pore water ⑨ Fissure water ⚲ Spring

Fig. 3. Hydrogeologic cross-section of the study area. Cross-section along Line 1 (Northwest-Southeast)

(a); cross-section along Line 2 (Southwest-Northeast) (b); location map of Line 1 and Line 2 within the study area (c). The Malan Loess (11.7–12.6 Ka BP) and Lishi Loess (12.6–78.1 Ka BP) are two major Quaternary loess stratigraphic units in China. Based on hydrogeological research, the stratigraphy of the hilly region features a multi-layer structure from top to bottom: Upper Pleistocene Malan Loess, Middle Pleistocene Lishi Loess, Neogene Red Clay and Mudstone (2.58–23.03 Ma BP), and Jurassic Sandstone and Conglomerate (145–201.3 Ma BP). In the gully region, the stratigraphy includes Holocene loess (silt, 11.7 ka BP–present), Middle Pleistocene Lishi Loess, Neogene sandstone and mudstone, and Jurassic sandstone and conglomerate, with some areas containing coal seams up to 5 meters thick.

2. The terms 'Pore water table' and 'Porous water table' in Fig. 9a should be standardized for consistency.

**Response:** We have addressed this comment by standardizing the terminology in Fig. 9a (in the original manuscript, and now Fig. 10a in the revised manuscript), consistently using "Pore water table" throughout. Additionally, we have reviewed the entire manuscript and made similar revisions to ensure consistency across all related expressions. The specific additions are as follows:

[Figure]

Fig. 10. Temporal dynamics of pore water table depth, temperature, precipitation, and recharge in the gully region of the Loess Plateau. (a) Daily time series of pore water table depth (blue line) and surface temperature (red line) from September 2023 to November 2024. The water table fluctuates seasonally, rising from ~−8.1 m in late summer to a maximum of ~−5.0 m in early spring (March 2024), indicating delayed infiltration and cool-season recharge. (b) Daily precipitation (blue bars) and modeled pore water recharge estimates using the MRC methods. (c) Daily precipitation (blue bars) and modeled pore water recharge estimates using the RISE methods. Most recharge events occur from October to April, even when rainfall is not especially high, while warm-season precipitation contributes little to recharge, likely due to increased evaporative losses and shallow soil retention. Together, these patterns suggest strong seasonal control on recharge processes, with effective infiltration primarily occurring during cooler, low-evaporation periods.

3. In Fig. 9b, the overlap of 'pore water recharge' and 'precipitation' affects the visibility of the recharge results. It is recommended to display precipitation separately or on the upper axis of the figure.

**Response:** In the original manuscript, we placed the precipitation and pore water recharge data on the same axis to better illustrate their synchronous relationship, which led to some visual overlap.

Following your comment, we have moved the precipitation data to the upper axis of Fig. 9b (in the original manuscript, and now Fig. 10b in the revised manuscript), ensuring that the visibility of the recharge results is not obstructed. The specific revision is as follows:

[Figure]

Fig. 10. Temporal dynamics of pore water table depth, temperature, precipitation, and recharge in the gully region of the Loess Plateau. (a) Daily time series of pore water table depth (blue line) and surface temperature (red line) from September 2023 to November 2024. The water table fluctuates seasonally, rising from ~−8.1 m in late summer to a maximum of ~−5.0 m in early spring (March

2024), indicating delayed infiltration and cool-season recharge. (b) Daily precipitation (blue bars)

and modeled pore water recharge estimates using the MRC methods. (c) Daily precipitation (blue bars) and modeled pore water recharge estimates using the RISE methods. Most recharge events occur from October to April, even when rainfall is not especially high, while warm-season precipitation contributes little to recharge, likely due to increased evaporative losses and shallow soil retention. Together, these patterns suggest strong seasonal control on recharge processes, with effective infiltration primarily occurring during cooler, low-evaporation periods.

---

## Author Comment (AC6)

**CC1**

Thank you for taking the time to review our manuscript again. We have thoroughly considered your feedback and made further refinements to the manuscript. **In response to your feedback and the comments from reviewers #RC1 and #RC2, your comments are presented in red font, our responses in black, and the revisions to the manuscript in blue. This is our reply to your second comments.**

The authors have thoroughly addressed all the reviewers' comments they have received, making detailed revisions, clarifications, and additional analyses. These changes notably improve the manuscript's clarity, rigor, and broader implications, especially in defining key mechanisms, addressing uncertainties, and incorporating relevant literature. I therefore recommend acceptance for publication in HESS with minor revisions.

**Response:** Thank you for your constructive feedback. The specific revisions are shown below.

1. There are too many dashes in the text; it is recommended to revise and simplify.

**Response:** Based on your comment, we have reviewed the entire manuscript and revised the sentences with dashed expressions.

2. I suggest replacing the keyword "connectivity" with "hydrological connectivity", and if possible add a keyword "groundwater recharge".

**Response:** We have revised and added keywords. The specific revisions are as follows:

"Keywords: surface water, spring water, pore water, fissure water, **hydrological connectivity, groundwater recharge**"

3. It is unclear why the legend colors in Figure 6 cover the text instead of being right-aligned.

**Response:** Thank you for your detailed comment regarding the format of Fig. 6. In the original manuscript, the legend (colors) in Fig. 6 represented both the x-axis categories of the $\delta^2H$ boxplot and the y-axis categories of the $\delta^{18}O$ boxplot, creating a composite legend. To avoid confusion, we have placed the explanatory text above the legend to clarify that it applies to both subplots. As suggested, we have also centered the text in Fig. 6 for better clarity. The specific revisions are as follows:

[Figure]

Fig. 6. Dual stable isotopic compositions of rainfall, pond water, spring water, pore water, and fissure water during the rainy season and dry season in the gully region of the Loess Plateau. The black line represents the global meteoric water line (GMWL, $\delta^2H=10+8\delta^{18}O$). GMWL is the global meteoric water line of Craig, LMWL is the local meteoric water line, SWL is the spring water line, POWL is the pond water line, FWL is the fissure water line, and PWL is the pore water line. Panels (b) and (d) are magnified views of (a) and (c), respectively, highlighting the isotopic compositions of pore water, fissure water, and spring water (x-axis: –12 to –6‰; y-axis: –80 to –

50‰).

4. Redrawing Figure 10 to include the main findings of this study would be beneficial.

**Response:** We have revised Fig. 10 (original manuscript, and now Fig. 11 in the revised manuscript) based on your comment, systematically integrating the main findings of this study into the conceptual model. Additionally, we have addressed the updates to Fig. 10 in the
discussion section to ensure consistency between the figure and the text. The specific revisions are
shown below:

[Figure]

Fig. 11. Hydraulic connections between different water bodies in the hilly-gully region of the
Loess Plateau. The study area consists of hilly and gully regions. In the hilly area, the stratigraphic
sequence from top to bottom is Malan loess, Lishi loess, red clay, sandstone, and mudstone.
Rainfall infiltration within the Malan loess is less than 1 m, and the area is mainly covered by
vegetation. In the gully area, the stratigraphy from top to bottom includes loess (silt), sandstone
and conglomerate, and mudstone. Pore water is found within the sandstone and conglomerate,
while fissure water occurs in bedrock fractures (mudstone). Numerous check dams or ponds are
distributed throughout the gully area. The vertical separation between the pore water and pond
water ranges from 3 to 5 m. Corn is the main crop cultivated in this region. Most springs in the
study area are located at the junction of the hilly and gully regions and are discharged from pore
water.